# Clean Images are Hard to Reblur: Exploiting the Ill-Posed Inverse Task for Dynamic Scene Deblurring

**Seungjun Nah**[1,2]*, **Sanghyun Son**[1], **Jaerin Lee**[1] **& Kyoung Mu Lee**[1]
[1]ASRI, Department of ECE, Seoul National University, Seoul, Korea    [2]NVIDIA
seungjun.nah@gmail.com, {thstkdgus35, ironjr, kyoungmu}@snu.ac.kr

## Abstract

The goal of dynamic scene deblurring is to remove the motion blur in a given image. Typical learning-based approaches implement their solutions by minimizing the L1 or L2 distance between the output and the reference sharp image. Recent attempts adopt visual recognition features in training to improve the perceptual quality. However, those features are primarily designed to capture high-level contexts rather than low-level structures such as blurriness. Instead, we propose a more direct way to make images sharper by exploiting the inverse task of deblurring, namely, reblurring. Reblurring amplifies the remaining blur to rebuild the original blur, however, a well-deblurred clean image with zero-magnitude blur is hard to reblur. Thus, we design two types of reblurring loss functions for better deblurring. The supervised reblurring loss at training stage compares the amplified blur between the deblurred and the sharp images. The self-supervised reblurring loss at inference stage inspects if there noticeable blur remains in the deblurred. Our experimental results on large-scale benchmarks and real images demonstrate the effectiveness of the reblurring losses in improving the perceptual quality of the deblurred images in terms of NIQE and LPIPS scores as well as visual sharpness.

## 1 Introduction

Motion blur commonly arises when the cameras move or scene changes during the exposure in dynamic environments. Dynamic scene deblurring is a challenging ill-posed task finding both the locally-varying blur and the latent sharp image from a large solution space. Traditional approaches (Hirsch et al., 2011; Whyte et al., 2012; Kim et al., 2013; Kim & Lee, 2014) tried to alleviate the ill-posedness by using statistical prior on sharp images such as gradient sparsity.

Instead of using such handcrafted knowledge, recent methods take advantage of large-scale datasets as well as deep neural networks (Nah et al., 2017; Su et al., 2017; Noroozi et al., 2017; Nah et al., 2019; Shen et al., 2019). Usually, the learning is driven by minimizing the pixel-wise distance to the ground truth, e.g., L1 or L2, so that the PSNR between the deblurred and the sharp reference can be maximized. By utilizing modern ConvNet architectures and training techniques, state-of-the-art approaches (Nah et al., 2017; Tao et al., 2017; Gao et al., 2019; Yuan et al., 2020; Park et al., 2020; Chi et al., 2021) have been developed toward higher capacity and deblurring accuracy. Still, most methods tend to suffer from the blurry predictions due to the regression-to-mean behavior often witnessed in ill-posed problems with large solution space (Ledig et al., 2017; Menon et al., 2020).

To overcome limitations of the conventional objectives, concepts of perceptual (Johnson et al., 2016) and adversarial (Ledig et al., 2017; Nah et al., 2017; Kupyn et al., 2018) loss terms from high-level semantic tasks have been introduced to improve the visual quality of the deblurred results. Nevertheless, such high-level losses may not serve as optimal goals for blur removal as low-level structural properties, e.g., blurriness, are not the primary features considered in their formulations. As illustrated in Figure 1, results from the previous deblurring methods are still blurry to a degree and the VGG and the adversarial losses are not sufficient to obtain perceptually pleasing and sharp images across different architectures (Tao et al., 2018; Gao et al., 2019; Kupyn et al., 2019).

---

*Most work was done at SNU

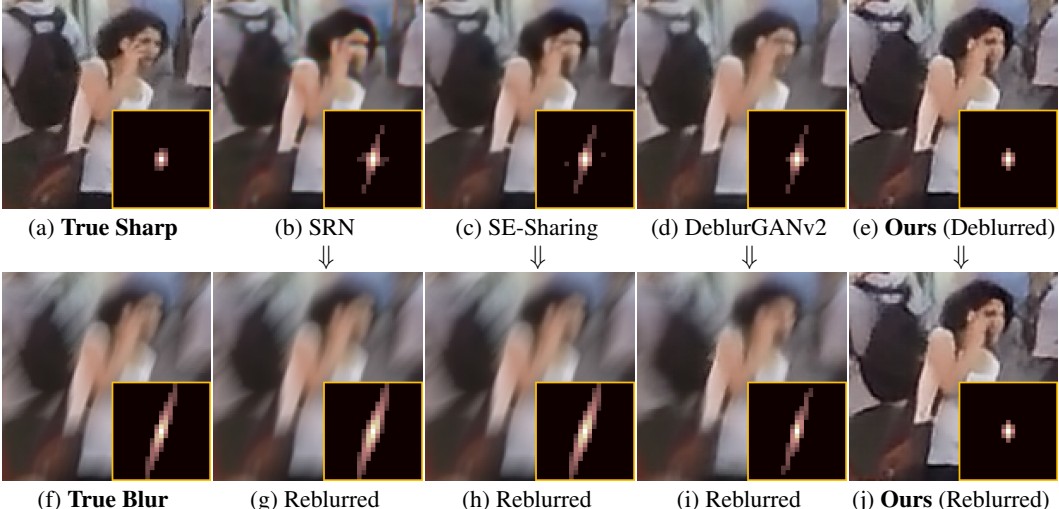

| (a) **True Sharp** | (b) SRN | (c) SE-Sharing | (d) DeblurGANv2 | (e) **Ours** (Deblurred) |
| | ⇓ | ⇓ | ⇓ | ⇓ |
| (f) **True Blur** | (g) Reblurred | (h) Reblurred | (i) Reblurred | (j) **Ours** (Reblurred) |

Figure 1: **Comparison of the deblurred images and their reblurred counterparts.** For each image, we visualize the remaining blur kernel (Cho & Lee, 2009) at the center pixel visualized on the right bottom side. **Upper**: The kernels from the previous deblurring methods implicate the direction of the original blur. **Lower**: When the proposed reblurring module is applied, our result does not lose sharpness as we reconstruct the output that is hard to be reblurred.

While the deblurred images look less blurry compared with the original input, it is still possible to find nontrivial blur kernels with directional motion information. From the observation, we introduce the concept of *reblurring* which amplifies the unremoved blur in the given image and reconstructs the original blur. We note that our reblurring operation aims to recover the original motion trajectory in the blurry input, rather than to synthesize arbitrary, e.g., Gaussian, blurs. Therefore, an ideally deblurred clean image is hard to reblur as no noticeable blur can be found to be amplified, making reblurring an ill-posed task. In contrast, it is straightforward to predict the original shape of blur from insufficiently deblurred images as shown in Figure 1. We propose to use the difference between non-ideally deblurred image and the ideal sharp image in terms of reblurring feasibility as the new optimization objective, *reblurring loss* for the image deblurring problem.

The reblurring loss is realized by jointly training a pair of deblurring and reblurring modules. The reblurring module performs the inverse operation of deblurring, trying to reconstruct the original blurry image from a deblurred output. Using the property that the blurriness of a reblurred image depends on the sharpness quality of the deblurred result, we construct two types of loss functions. During the joint training, *supervised reblurring loss* compares the amplified blurs between the deblurred and the sharp image. Complementing L1 intensity loss, the supervised reblurring loss guides the deblurring module to focus on and eliminate the remaining blur. While our training strategy is similar to the adversarial training of GANs (Goodfellow et al., 2014) in a sense that our deblurring and reblurring modules play the opposite roles, the purposes and effects of the adversary are different. The reblurring loss concentrates on image blurriness regardless of image realism. Furthermore, in contrast to the GAN discriminators that are not often used at test time, our reblurring module can be used to facilitate *self-supervised reblurring loss*. By making the deblurred image harder to reblur, the deblurring module can adaptively optimize itself without referring to the ground truth.

Our reblurring loss functions provide additional optimization directives to the deblurring module and can be generally applied to any learning-based image deblurring methods. With the proposed approach, we can derive sharper predictions from existing deblurring methods without modifying their architectures. We summarize our contributions as follows:

- Based on the observation that clean images are hard to reblur, we propose novel loss functions for image deblurring. Our reblurring loss reflects the preference for sharper images and contributes to visually pleasing deblurring results.
- At test-time, the reblurring loss can be implemented without a ground-truth image. We perform test-time adaptive inference via self-supervised optimization with each input.
- Our method is generally applicable to any learning-based methods and jointly with other loss terms. Experiments show that the concept of reblurring loss consistently contributes to achieving state-of-the-art visual sharpness as well as LPIPS and NIQE across different model architectures.

## 2 RELATED WORKS

**Image Deblurring.** Classical energy optimization framework is formulated by likelihood and prior terms. Due to the ill-posedness of dynamic scene deblurring problem, prior terms have been essential in alleviating the optimization ambiguity, encoding the preference on the solutions. Sophisticated prior terms were carefully designed with human knowledge on natural image statistics (Levin, 2006; Cho & Lee, 2009; Hirsch et al., 2011; Whyte et al., 2012; Sun et al., 2013; Xu et al., 2013; Kim et al., 2013; Kim & Lee, 2014; Pan et al., 2016). Recently in Li et al. (2018), learned prior from a classifier discriminating blurry and clean images was also shown to be effective. Deep priors were also used for image deconvolution problems (Ren et al., 2020; Nan & Ji, 2020).

On the other hand, deep learning methods have benefited from learning on large-scale datasets. The datasets consisting of realistic blur (Nah et al., 2017; Su et al., 2017; Noroozi et al., 2017; Nah et al., 2019; Gao et al., 2019; Jin et al., 2019; Shen et al., 2019) align the temporal center of the blurry and the sharp image pairs with high-speed cameras. Learning from such temporally aligned data relieve the ill-posedness of deblurring compared with difficult energy optimization framework. Thus, more attention has been paid to designing CNN architectures and datasets than designing loss terms.

In the early work of Schuler et al. (2015), the alternating estimation of blur kernel and restored image (Cho & Lee, 2009) was adopted in CNN architecture. In Sun et al. (2015); Gong et al. (2017), the spatially varying blur kernels are estimated by assuming locally linear blur followed by non-blind deconvolution. Later, end-to-end learning without explicit kernel estimation became popular. Motivated from the coarse-to-fine approach, multi-scale CNN was proposed (Nah et al., 2017) to expand the receptive field efficiently, followed by scale-recurrent architectures (Tao et al., 2018; Gao et al., 2019). On the other hand, Zhang et al. (2019); Suin et al. (2020) sequentially stacked network modules. Recently, Park et al. (2020) proposed a multi-temporal model that deblurs an image recursively. To handle spatially varying blur kernels efficiently, spatially non-uniform operations were embedded in neural networks (Zhang et al., 2018a; Yuan et al., 2020).

**Perceptual Image Restoration.** Often, L1 or L2 losses are used at training to achieve higher PSNR. However, such approaches suffer from blurry and over-smoothed outputs (Johnson et al., 2016; Zhang et al., 2018b; Menon et al., 2020) as the learned models predict an average of all possible solutions under the ill-posedness (Ledig et al., 2017). To deal with the issue, several studies utilize deep features of the pretrained VGG (Simonyan & Zisserman, 2014) and other networks that are more related to human perception (Johnson et al., 2016; Zhang et al., 2018b) and with analysis on frequency space (Tariq et al., 2020; Czolbe et al., 2020). Recent methods introduce adversarial training (Goodfellow et al., 2014) so that outputs of the restoration models be indistinguishable from real samples (Nah et al., 2017; Nimisha et al., 2017; Ledig et al., 2017; Kupyn et al., 2018; 2019). Also, there were attempts to exploit statistical properties of images and features with contextual loss (Mechrez et al., 2018) and projected distribution loss (Delbracio et al., 2021).

Nevertheless, an inherent limitation of existing perceptual objectives is that they are not task-specialized for image restoration. For example, the VGG features are learned for high-level visual recognition while the adversarial loss only contributes to reconstructing realistic images without considering the existence of motion blur. Therefore, blindly optimizing those terms may not yield an optimal solution in terms of image deblurring. In practice, we observed that those objectives still tend to leave blur footprints unremoved, making it possible to estimate the original blur. Our reblurring loss is explicitly designed to improve the perceptual sharpness of deblurred images by reducing remaining blurriness and thus more suitable for deblurring, acting as a learned prior.

**Image Blurring.** As an image could be blurred in various directions and strength, image blurring is another ill-posed problem without additional information. Thus, intrinsic or extrinsic information is often incorporated. With a non-ideally sharp image, Bae & Durand (2007) detected the small local blur kernel in the image to magnify the defocus blur for bokeh effect. On the other hand, Chen et al. (2018) estimated the kernel by computing the optical flow from the neighboring video frames. Similarly, Brooks & Barron (2019) used multiple video frames to synthesize blur. Without such external information, Zhang et al. (2020) used a generative model to synthesize many blurry images. In contrast, Bahat et al. (2017) deliberately blurred an already blurry image in many ways to find the local blur kernel. Our image reblurring concept is similar to Bae & Durand (2007) in the sense that intrinsic cue in an image is used to amplify blur. Nonetheless, our main goal is to use reblurring to provide a guide to deblurring model so that such blur cues would be better removed.

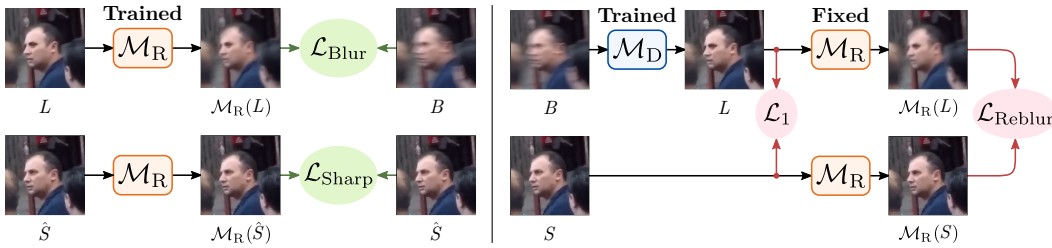

(a) Reblurring module training process.  (b) Image deblurring with reblurring loss

Figure 2: **Overviews of the proposed reblurring and deblurring framework.** Reblurring module $\mathcal{M}_R$ tries to reconstruct blurry image $B$ from a deblurred image $L$ while preserving the sharpness of a pseudo-sharp image $\hat{S} = \mathcal{M}_D(S)$. Meanwhile, the deblurring module $\mathcal{M}_D$ tries to make $L$ sharper by comparing the amplified blur from $L$ and the sharp image $S$.

| #ResBlocks in deblurring module $\mathcal{M}_D$ | 4 | 8 | 16 | 32 |
|---|---|---|---|---|
| Deblur PSNR wrt sharp GT | 28.17 | 29.67 | 30.78 | 31.48 |
| Reblur PSNR wrt blur GT | 34.29 | 32.66 | 31.90 | 31.48 |

Table 1: **Deblurring and reblurring PSNR (dB) by deblurring model capacity.** Both tasks are trained independently with L1 loss on the GOPRO dataset. The number of ResBlocks in $\mathcal{M}_R$ is 2.

## 3  PROPOSED METHOD

In this section, we describe a detailed concept of image reblurring and how the operation can be learned. Then, we demonstrate that the operation can support the deblurring module to reconstruct perceptually favorable and sharp images. We also propose a self-supervised test-time optimization strategy by using the learned reblurring module. For simplicity, we refer to the blurry, the deblurred, and the sharp image as $B$, $L$, and $S$, respectively.

### 3.1  CLEAN IMAGES ARE HARD TO REBLUR

As shown in Figure 1, outputs from the existing deblurring methods still contain undesired motion trajectories that are not completely removed from the input. Ideally, a well-deblurred image should not contain any motion cues, making reblurring ill-posed. We first validate our motivation by buliding a reblurring module $\mathcal{M}_R$ which amplifies the remaining blur from $L$. $\mathcal{M}_R$ is trained with the following blur reconstruction loss $\mathcal{L}_{Blur}$ so that it would learn the inverse operation of deblurring.

$$\mathcal{L}_{Blur} = \|\mathcal{M}_R(L) - B\|. \tag{1}$$

We apply $\mathcal{M}_R$ to the deblurred images from deblurring modules of varying capacities. Table 1 shows that the higher the deblurring PSNR, the lower the reblurring PSNR becomes when both modules are trained with conventional L1 loss, independently from each other. It demonstrates that the better deblurred images are harder to reblur, consistent to our motivation.

In contrast to the non-ideally deblurred images, $\mathcal{M}_R$ is not able to generate a motion blur from a sharp image $S$ as no motion information is found. For a high-quality clean image, $\mathcal{M}_R$ should preserve the sharpness. However, optimizing the blur reconstruction loss $\mathcal{L}_{Blur}$ with $S$ may fall into learning the pixel average of all blur trajectories in the training dataset, i.e. Gaussian blur. In such a case, $\mathcal{M}_R$ will apply the single uniform blur on every image without considering the scene information. To let the blur domain of $\mathcal{M}_R$ confined to the motion-incurred blur, we use sharp images to penalize such undesired operations. Specifically, we introduce a network-generated sharp image $\hat{S}$ obtained by feeding a real sharp image $S$ to the deblurring module $\mathcal{M}_D$ as $\hat{S} = \mathcal{M}_D(S)$. We define sharpness preservation loss $\mathcal{L}_{Sharp}$ as follows:

$$\mathcal{L}_{Sharp} = \|\mathcal{M}_R(\hat{S}) - \hat{S}\|. \tag{2}$$

We use the pseudo-sharp image $\hat{S}$ instead of a real image $S$ to make our reblurring module focus on image sharpness and blurriness rather than the realism. While $\hat{S}$ differ from $L$ only by the sharpness, $S$ also differ by the realism which can be easily detected by neural networks (Wang et al., 2020).

Combining both terms together, we train the reblurring module $\mathcal{M}_R$ by optimizing the joint loss $\mathcal{L}_R$:

$$\mathcal{L}_R = \mathcal{L}_{Blur} + \mathcal{L}_{Sharp}. \tag{3}$$

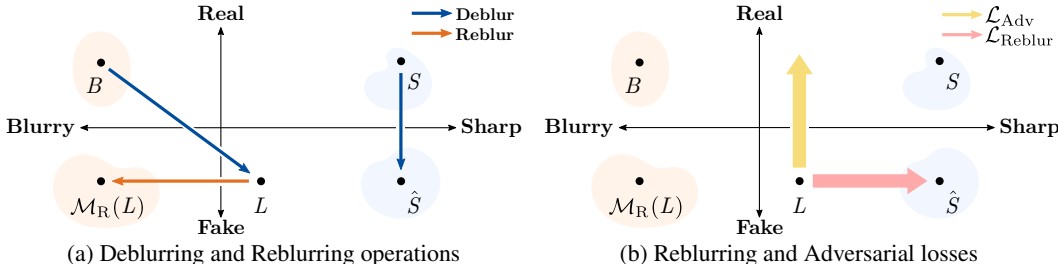

(a) Deblurring and Reblurring operations     (b) Reblurring and Adversarial losses

Figure 3: **Image deblurring and reblurring illustrated from the perspective of sharpness and realism.** Training our modules with $\mathcal{L}_{\text{Reblur}}$ improves image sharpness without considering the image realism. The image realism can be optionally handled by adversarial loss $\mathcal{L}_{\text{Adv}}$.

As zero-magnitude blur should remain unaltered from $\mathcal{M}_{\text{R}}$, the sharpness preservation loss can be considered a special case of the blur reconstruction loss. Figure 2a illustrates the way our reblurring module is trained from $\mathcal{L}_{\text{R}}$.

### 3.2 Supervision from Reblurring Loss

The blurriness of images can be more easily witnessed by amplifying the blur. Thus, we propose a new optimization objective by processing the deblurred and the sharp image with the jointly trained reblurring model $\mathcal{M}_{\text{R}}$. To suppress the remaining blur in the output image $L = \mathcal{M}_{\text{D}}(B)$ from the deblurring module $\mathcal{M}_{\text{D}}$, the *supervised reblurring loss* $\mathcal{L}_{\text{Reblur}}$ for image deblurring is defined as

$$\mathcal{L}_{\text{Reblur}} = \|\mathcal{M}_{\text{R}}(L) - \mathcal{M}_{\text{R}}(S)\|. \tag{4}$$

Unlike the sharpness preservation term in equation 2, we do not use the pseudo-sharp image $\hat{S}$ in our reblurring loss, $\mathcal{L}_{\text{Reblur}}$. As the quality of the pseudo-sharp image $\hat{S}$ depends on the state of deblurring module $\mathcal{M}_{\text{D}}$, using $\hat{S}$ may make training unstable and difficult to optimize, especially at the early stage. Thus we use a real sharp image $S$ to stabilize the training. Nevertheless, as $\mathcal{M}_{\text{R}}$ is trained to focus on the sharpness from equation 3, so does the reblurring loss, $\mathcal{L}_{\text{Reblur}}$.

Using our reblurring loss in equation 4, the deblurring module $\mathcal{M}_{\text{D}}$ is trained to minimize the following objective $\mathcal{L}_{\text{D}}$:

$$\mathcal{L}_{\text{D}} = \mathcal{L}_1 + \lambda \mathcal{L}_{\text{Reblur}}, \tag{5}$$

where $\mathcal{L}_1$ is a conventional L1 loss, and the hyperparameter $\lambda$ is empirically set to 1. Figure 2b illustrates how the deblurring model is trained with the guide from $\mathcal{M}_{\text{R}}$.

At each training iterations, we alternately optimize two modules $\mathcal{M}_{\text{D}}$ and $\mathcal{M}_{\text{R}}$ by $\mathcal{L}_{\text{D}}$ and $\mathcal{L}_{\text{R}}$, respectively. While such a strategy may look similar to the adversarial training scheme, the optimization objectives are different. As the neural networks are well known to easily discriminate real and fake images (Wang et al., 2020), the realism could be as a more obvious feature than image blurriness. Thus, adversarial loss may overlook image blurriness as $L$ and $S$ can already be discriminated by the difference in realism. On the other hand, our reblurring loss is explicitly designed to prefer sharp images regardless of realism as we use $\hat{S}$ instead of $S$ in $\mathcal{L}_{\text{Sharp}}$ to train $\mathcal{M}_{\text{R}}$. Figure 3 conceptually compares the actual role of the reblurring loss $\mathcal{L}_{\text{Reblur}}$ and the adversarial loss $\mathcal{L}_{\text{Adv}}$.

### 3.3 Test-time Adaptation by Self-Supervision

After the training is over, the models learned from supervised loss terms have fixed weights at test time. When a new example that deviates from the distribution of training data is given, the supervised methods may lack ability to generalize. Our reblurring module, however, can further provide self-supervised guide so that the model could be further optimized for each image at test time. While the supervised reblurring loss $\mathcal{L}_{\text{Reblur}}$ finds the blurriness of $L$ by comparison with the ground truth, $\mathcal{M}_{\text{R}}$ can also inspect the blurriness of an image without reference.

As $\mathcal{M}_{\text{R}}$ is trained to magnify the blur in $L$, imperfectly deblurred image would be blurred. Thus, the difference between $\mathcal{M}_{\text{R}}(L)$ and $L$ can serve as a feedback without having to reference $S$. Furthermore, due to the sharpness preservation loss $\mathcal{L}_{\text{Sharp}}$, a sufficiently sharp image would have little difference when reblurred. Based on the property, we construct the *self-supervised reblurring loss*

| Model | Optimization | On GOPRO dataset | | | | On REDS dataset | | | |
|---|---|---|---|---|---|---|---|---|---|
| | | LPIPS$_\downarrow$ | NIQE$_\downarrow$ | PSNR$^\uparrow$ | SSIM$^\uparrow$ | LPIPS$_\downarrow$ | NIQE$_\downarrow$ | PSNR$^\uparrow$ | SSIM$^\uparrow$ |
| U-Net | $\mathcal{L}_1$ | 0.1635 | 5.996 | 29.66 | 0.8874 | 0.1486 | 3.649 | 30.80 | 0.8772 |
| | $\mathcal{L}_1 + \mathcal{L}_{\text{Reblur, n1}}$ | 0.1365 | 5.629 | 29.58 | 0.8869 | 0.1435 | 3.487 | 30.76 | 0.8776 |
| | + TTA step 5 | **0.1327** | **5.599** | 29.52 | 0.8878 | **0.1403** | **3.476** | 30.64 | 0.8781 |
| | $\mathcal{L}_1 + \mathcal{L}_{\text{Reblur, n2}}$ | 0.1238 | 5.124 | 29.44 | 0.8824 | 0.1252 | 2.918 | 30.46 | 0.8717 |
| | + TTA step 5 | **0.1187** | **5.000** | 29.42 | 0.8831 | **0.1226** | **2.849** | 30.25 | 0.8701 |
| SRN | $\mathcal{L}_1$ | 0.1246 | 5.252 | 30.62 | 0.9078 | 0.1148 | 3.392 | 31.89 | 0.8999 |
| | $\mathcal{L}_1 + \mathcal{L}_{\text{Reblur, n1}}$ | 0.1140 | 5.136 | 30.74 | 0.9104 | 0.1071 | 3.305 | 32.01 | 0.9044 |
| | + TTA step 5 | **0.1101** | **5.079** | 30.60 | 0.9100 | **0.1029** | **3.278** | 31.83 | 0.9040 |
| | $\mathcal{L}_1 + \mathcal{L}_{\text{Reblur, n2}}$ | 0.1037 | 4.887 | 30.57 | 0.9074 | 0.0947 | 2.875 | 31.82 | 0.9026 |
| | + TTA step 5 | **0.0983** | **4.730** | 30.44 | 0.9067 | **0.0909** | **2.798** | 31.50 | 0.9008 |
| DHN | $\mathcal{L}_1$ | 0.1179 | 5.490 | 31.53 | 0.9207 | 0.0942 | 3.288 | 32.65 | 0.9152 |
| | $\mathcal{L}_1 + \mathcal{L}_{\text{Reblur, n1}}$ | 0.0975 | 5.472 | 31.53 | 0.9217 | 0.0931 | 3.248 | 32.57 | 0.9143 |
| | + TTA step 5 | **0.0940** | **5.343** | 31.32 | 0.9208 | **0.0887** | **3.220** | 32.38 | 0.9139 |
| | $\mathcal{L}_1 + \mathcal{L}_{\text{Reblur, n2}}$ | 0.0837 | 5.076 | 31.34 | 0.9177 | 0.0805 | 2.830 | 32.44 | 0.9122 |
| | + TTA step 5 | **0.0805** | **4.948** | 31.28 | 0.9174 | **0.0763** | **2.761** | 32.17 | 0.9110 |

Table 2: **Quantitative analysis of the reblurring losses and test-time adaptation applied to various deblurring networks on GOPRO and REDS datasets.**

that serves as a prior term embedding the preference on sharp images as

$$\mathcal{L}_{\text{Reblur}}^{\text{self}} = \|\mathcal{M}_{\text{R}}(L) - L_*\|, \tag{6}$$

where $L_*$ denotes the image with the same value as $L$ but the gradient does not backpropagate in the optimization process. We minimize $\mathcal{L}_{\text{Reblur}}^{\text{self}}$ for each test data to obtain the sharper image. Allowing gradient to flow through $L_*$ can let $L$ to fall into undesired local minima where both the $L$ and $\mathcal{M}_{\text{R}}(L)$ are blurry. We iteratively optimize the weights of $\mathcal{M}_{\text{D}}$ with fixed $\mathcal{M}_{\text{R}}$. As $\mathcal{L}_{\text{Reblur}}^{\text{self}}$ only considers the sharpness of an image, we keep the color consistency by matching the color histogram between the test-time adapted image and the initially deblurred image. For the detailed optimization process of test-time adaptation strategy, please refer to the Appendix Algorithm A. The effect of test-time adaptation is conceptually visualized in Figure 5. More iterations make the deblurred image sharper. We note that our loss functions and the test-time adaptation are applicable to general learning-based approaches.

## 4 EXPERIMENTS

We verify the effectiveness of our reblurring loss and the generalization by applying it to multiple deblurring model architectures. We show the experimental results with a baseline residual U-Net and state-of-the-art image deblurring models, the sRGB version SRN (Tao et al., 2018) and DHN, our modified version of DMPHN (Zhang et al., 2019). For the reblurring module, we use simple residual networks with 1 or 2 ResBlock(s) with $5 \times 5$ convolution kernels. The training and evaluation were done with the widely used GOPRO (Nah et al., 2017) and REDS (Nah et al., 2019) datasets. The GOPRO dataset consists of 2103 training and 1111 test images with various dynamic motion blur. Similarly, the REDS dataset has 24000 training and 3000 validation data publicly available. For each dataset, every experiment was done in the same training environment. We mainly compare LPIPS (Zhang et al., 2018b) and NIQE (Mittal et al., 2012) perceptual metrics as our goal is to make images sharper. For more implementation details, please refer to the Appendix.

### 4.1 EFFECT OF SUPERVISED REBLURRING LOSS

We implement the reblurring loss in varying degrees of emphasis on sharpness by controlling the reblurring module capacity. For a more balanced quality between PSNR and perceptual sharpness, we use 1 ResBlock for $\mathcal{M}_{\text{R}}$. To put more weight on the perceptual quality, we allocate a larger capacity on $\mathcal{M}_{\text{R}}$ by using 2 ResBlocks. For notation simplicity, we denote the reblurring loss with $k$ ResBlock(s) in the reblurring module as $\mathcal{L}_{\text{Reblur, n}k}$.

Table 2 shows how the deblurring performance varies by the use of reblurring loss functions. With $\mathcal{L}_{\text{Reblur, n1}}$, LPIPS and NIQE improves to a moderate degree while PSNR and SSIM metrics remain

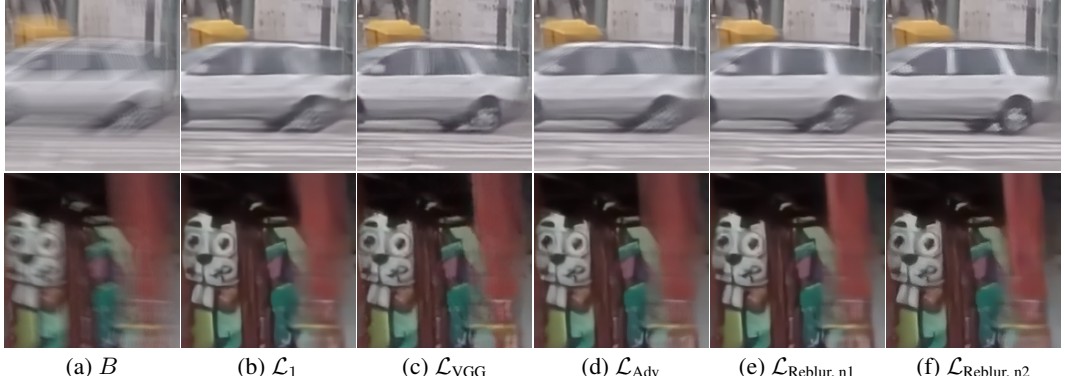

| (a) $B$ | (b) $\mathcal{L}_1$ | (c) $\mathcal{L}_{\text{VGG}}$ | (d) $\mathcal{L}_{\text{Adv}}$ | (e) $\mathcal{L}_{\text{Reblur, n1}}$ | (f) $\mathcal{L}_{\text{Reblur, n2}}$ |

Figure 4: **Visual comparison of deblurred results by training loss function on GOPRO dataset. Upper:** SRN, **Lower:** U-Net.

| Method | LPIPS$_\downarrow$ | NIQE$_\downarrow$ | PSNR$^\uparrow$ | SSIM$^\uparrow$ | Method | LPIPS$_\downarrow$ | NIQE$_\downarrow$ | PSNR$^\uparrow$ | SSIM$^\uparrow$ |
|---|---|---|---|---|---|---|---|---|---|
| U-Net (baseline) | 0.1635 | 5.996 | 29.66 | 0.8874 | SRN ($\mathcal{L}_1$) | 0.1246 | 5.252 | 30.62 | 0.9078 |
| U-Net ($\mathcal{L}_{\text{Blur}}$) | 0.1301 | 5.132 | 29.47 | 0.8839 | $+0.001\mathcal{L}_{\text{Adv}}$ | 0.1141 | 4.960 | 30.53 | 0.9068 |
| $+\mathcal{L}_{\text{Sharp}}$ with $S$ | 0.1410 | 5.307 | 29.15 | 0.8694 | $+0.3\mathcal{L}_{\text{VGG}}$ | **0.1037** | 4.945 | 30.60 | 0.9074 |
| $+\mathcal{L}_{\text{Sharp}}$ with $\hat{S}$ | **0.1238** | **5.124** | 29.44 | 0.8824 | $+\mathcal{L}_{\text{Reblur, n2}}$ | **0.1037** | **4.887** | 30.57 | 0.9074 |

Table 3: **The effect of the sharpness preservation in training our reblurring module measured on GOPRO dataset.**

Table 4: **Comparison of reblurring loss and other perceptual losses on GOPRO dataset applied to SRN.**

at a similar level. Meanwhile, $\mathcal{L}_{\text{Reblur, n2}}$ more aggressively optimizes the perceptual metrics. This is analogous to the perception-distortion trade-off witnessed in the image restoration literature (Blau & Michaeli, 2018; Blau et al., 2018). The perceptual metric improvements are consistently witnessed with various architectures on both the GOPRO and the REDS dataset.

## 4.2 Effect of Sharpness Preservation Loss

In training $\mathcal{M}_R$, we used both the blur reconstruction loss $\mathcal{L}_{\text{Blur}}$ and the sharpness preservation loss $\mathcal{L}_{\text{Sharp}}$. The latter term $\mathcal{L}_{\text{Sharp}}$ plays an essential role in letting $\mathcal{M}_R$ concentrate only on the motion-driven blur in the given image and keep sharp image remain sharp. Table 3 presents the performance gains from using $\mathcal{L}_{\text{Sharp}}$ jointly with $\mathcal{L}_{\text{Blur}}$ in terms of the perceptual quality.

Also, the use of the pseudo-sharp image $\hat{S}$ is justified by comparing it with the use case of real sharp image $S$. We found the using $S$ for $\mathcal{L}_{\text{Sharp}}$ with $\mathcal{L}_{\text{Blur}}$ makes the training less stable than using $\hat{S}$. Using the pseudo-sharp image confines the input data distribution of $\mathcal{M}_R$ to the domain of $\mathcal{M}_D$ outputs. For neural networks, it is very easy to discriminate real and fake images (Wang et al., 2020). By using a fake image $\hat{S}$ instead of a real image $S$, we let $\mathcal{M}_R$ focus on the sharpness of an image and avoid being distracted by a more obvious difference between real and fake images. Furthermore, it leads the two loss terms $\mathcal{L}_{\text{Blur}}$ and $\mathcal{L}_{\text{Sharp}}$ to reside under the same objective, amplifying any noticeable blur and keeping sharpness when motion blur is in zero-magnitude.

## 4.3 Comparison with Other Perceptual Losses

The reblurring loss provides a conceptually different learning objectives from the adversarial and the perceptual losses and is designed to focus on the motion blur. Table 4 compares the effect of $\mathcal{L}_{\text{Reblur}}$ with adversarial loss $\mathcal{L}_{\text{Adv}}$, and the VGG perceptual loss (Johnson et al., 2016) by applying them to SRN (Tao et al., 2018) on GOPRO dataset. While our method provides quantitatively better perceptual scores, the different perceptual losses are oriented to different goals. They do not necessarily compete or conflict with each other and can be jointly applied at training to catch the perceptual quality in varying aspects. In Figure 4, the effect of the reblurring loss is visually compared with the previous perceptual loss functions.

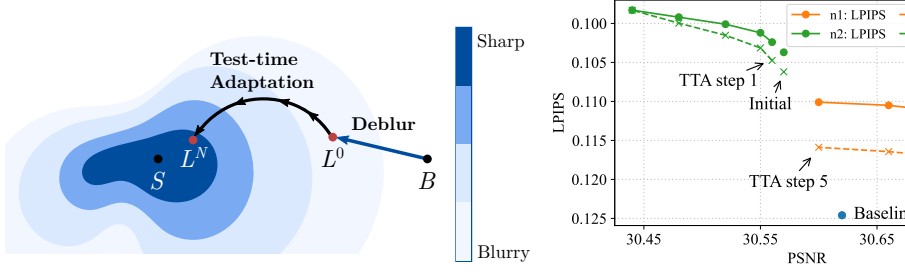

Figure 5: **The proposed self-supervised test-time adaptation.** The iterative optimization improves the image sharpness by finding an image that reblurs to the current deblurred image.

Figure 6: **Test-time adaption (SRN) on GOPRO dataset.** Reblurring loss improves the trade-off between the perception (LPIPS, NIQE) and PSNR compared with the baseline.

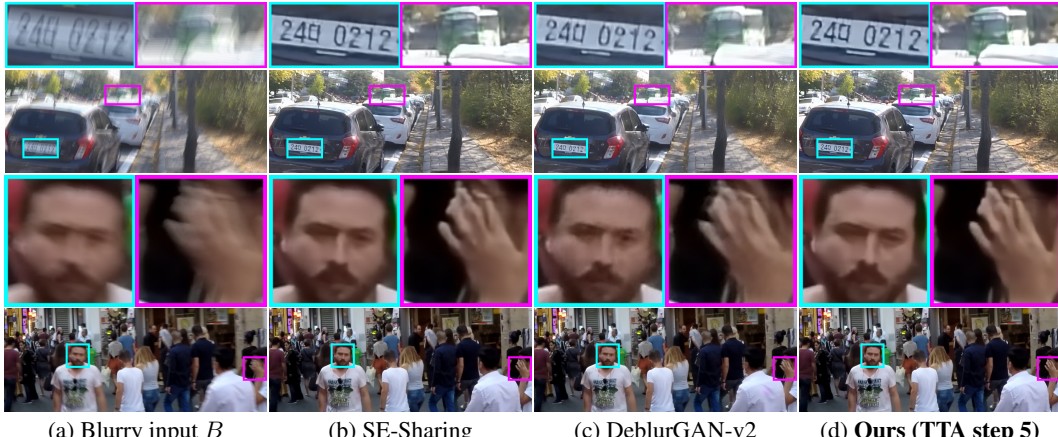

| (a) Blurry input $B$ | (b) SE-Sharing | (c) DeblurGAN-v2 | (d) **Ours (TTA step 5)** |

Figure 7: **Qualitative comparison between state-of-the-art deblurring methods on the GOPRO dataset.** We used the SRN model as a baseline architecture.

### 4.4 EFFECT OF TEST-TIME ADAPTATION

We conduct test-time adaptation with the proposed self-supervised reblurring loss, $\mathcal{L}_{\text{Reblur}}^{\text{self}}$ to make the deblurred image even sharper. Figure 6 shows the test-time-adapted result with SRN. Compared with the baseline trained with L1 loss, our results exhibit improved trade-off relations between PSNR and the perceptual metrics, LPIPS and NIQE. Table 2 provides detailed quantitative test-time adaptation results on GOPRO and REDS dataset, respectively with various deblurring module architectures. The effect of test-time adaptation is visually shown in Figure 8.

### 4.5 COMPARISON WITH STATE-OF-THE-ART METHODS

We have improved the perceptual quality of the deblurred images by training several different model architectures. We compare the perceptual quality with the other state-of-the-art methods in Figure 7. Especially, DeblurGAN-v2 was trained with the VGG loss and the adversarial loss. Our results achieve visually sharper texture from the reblurring loss and test-time adaptation.

### 4.6 REAL WORLD IMAGE DEBLURRING

While our method uses synthetic datasets (Nah et al., 2017; 2019) for training, the trained models generalize to real blurry images. In Figure 9, we show deblurred results from Lai et al. (2016) dataset with DHN model. Compared with the baseline $\mathcal{L}_1$ loss, our reblurring loss $\mathcal{L}_{\text{Reblur, n2}}$ provides an improved deblurring quality. As the real test image could deviate from the training data distribution, a single forward inference may not produce optimal results. With the self-supervised test-time adaptation, our deblurred images reveal sharper and detailed textures.

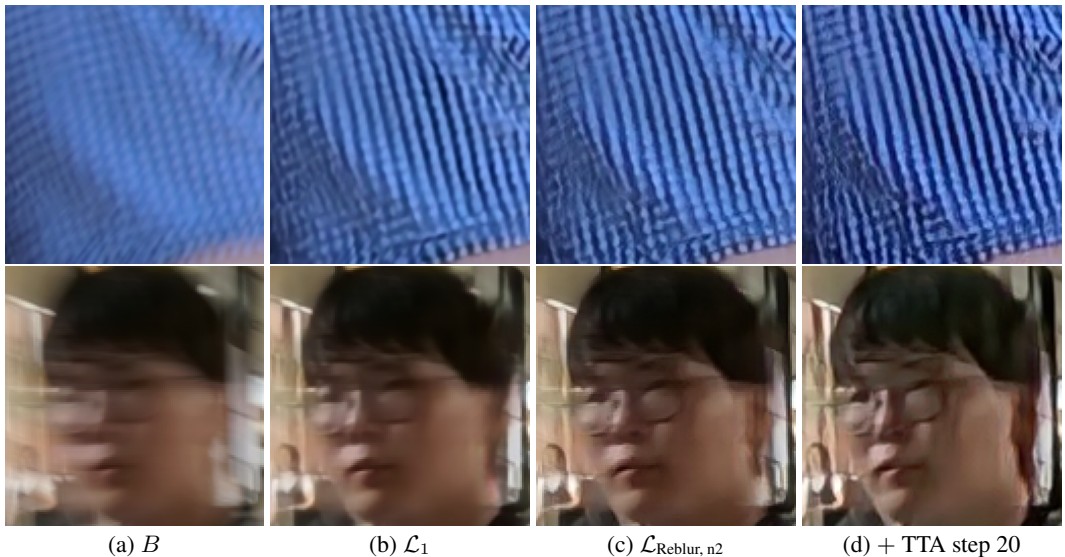

| (a) $B$ | (b) $\mathcal{L}_1$ | (c) $\mathcal{L}_{\text{Reblur, n2}}$ | (d) + TTA step 20 |

Figure 8: **Qualitative comparison between different training objectives and the test-time adaptation.** Patches are sampled from the REDS dataset validation set.

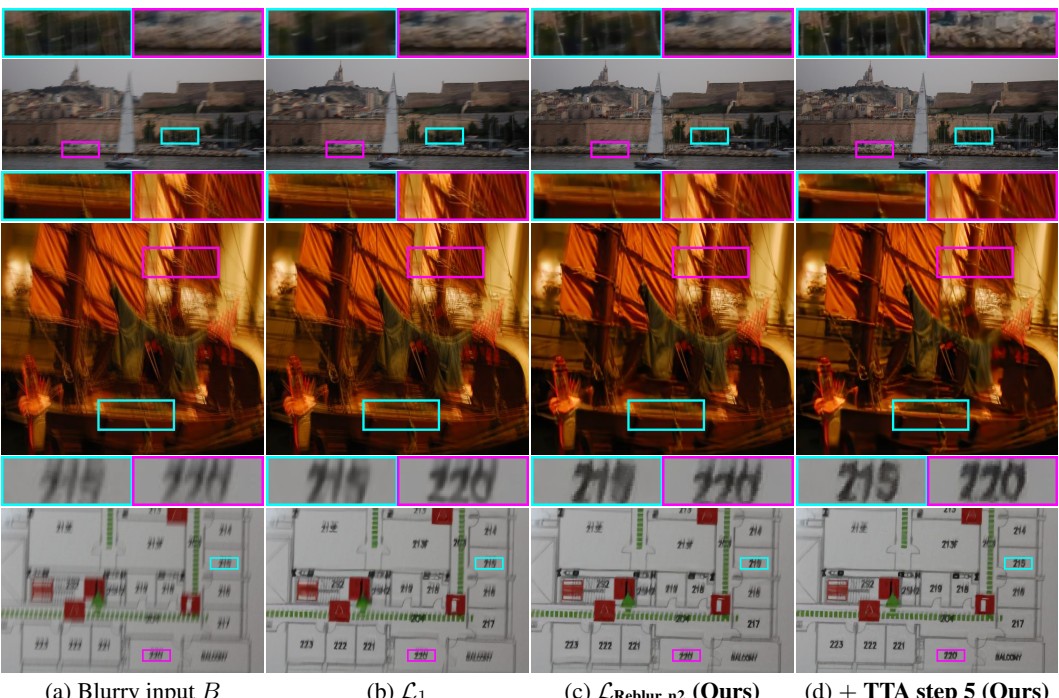

| (a) Blurry input $B$ | (b) $\mathcal{L}_1$ | (c) $\mathcal{L}_{\text{Reblur, n2}}$ **(Ours)** | (d) + **TTA step 5 (Ours)** |

Figure 9: **Qualitative comparison of deblurring results on the real-world images (Lai et al., 2016) by different loss functions and test-time adaptation.** The proposed test-time adaptation greatly improves visual quality and sharpness of the deblurred images.

## 5 CONCLUSION

In this paper, we validate a new observation that clean sharp images are hard to reblur and develop novel low-level perceptual objective terms, namely reblurring loss. The term is constructed to care the image blurriness by jointly training a pair of deblurring and reblurring modules. The supervised reblurring loss provides an amplified comparison on motion blur while the self-supervised loss inspects the blurriness in a single image with the learned reblurring module. The self-supervision lets the deblurring module adapt to the new image at test time without ground truth. By applying the loss terms to state-of-the-art deblurring methods, we demonstrate our method consistently improves the the perceptual sharpness of the deblurred images visually as well as quantitatively.

ACKNOWLEDGMENTS

This work was supported in part by IITP grant funded by the Korea government [No. 2021-0-01343, Artificial Intelligence Graduate School Program (Seoul National University)], and in part by AIRS Company in Hyundai Motor Company & Kia Motors Corporation through HMC/KIA-SNU AI Consortium.

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

## A  APPENDIX

In this appendix, we provide the implementation details and additional experimental analysis. In Section B, we explain the implementation details with the model architecture specifics, training details, and the evaluation metrics. Section C describes how the reblurring module design and the size are determined. Then in Section D, we describe the different characteristics of the proposed reblurring loss and the other perceptual losses. We combine our reblurring loss with the other perceptual losses to take advantage in multiple perspectives. In Section B, the test-time adaptation algorithm is described. In Section F, we show the quantitative effect of test-time adaptation and show the trade-off relation between the conventional distortion quality metric (PSNR, SSIM) and the perceptual metrics (LPIPS, NIQE) compared with the baselines.

## B  IMPLEMENTATION DETAILS

**Model Architecture.** In the main manuscript, we mainly performed the experiments with 3 different model architectures. First, we set our baseline model as a light-weight residual U-Net architecture that runs in a fast speed. The baseline model is used to design our reblurring loss with pseudo-sharp images through ablation study in Table 3.

For reblurring operation, we use a simple residual network $\mathcal{M}_R$ without strides to avoid deconvolution artifacts. The baseline U-Net and the reblurring module architectures are shown in Figure A. The detailed parameters for U-Net and $\mathcal{M}_R$ are each specified in Table A and C.

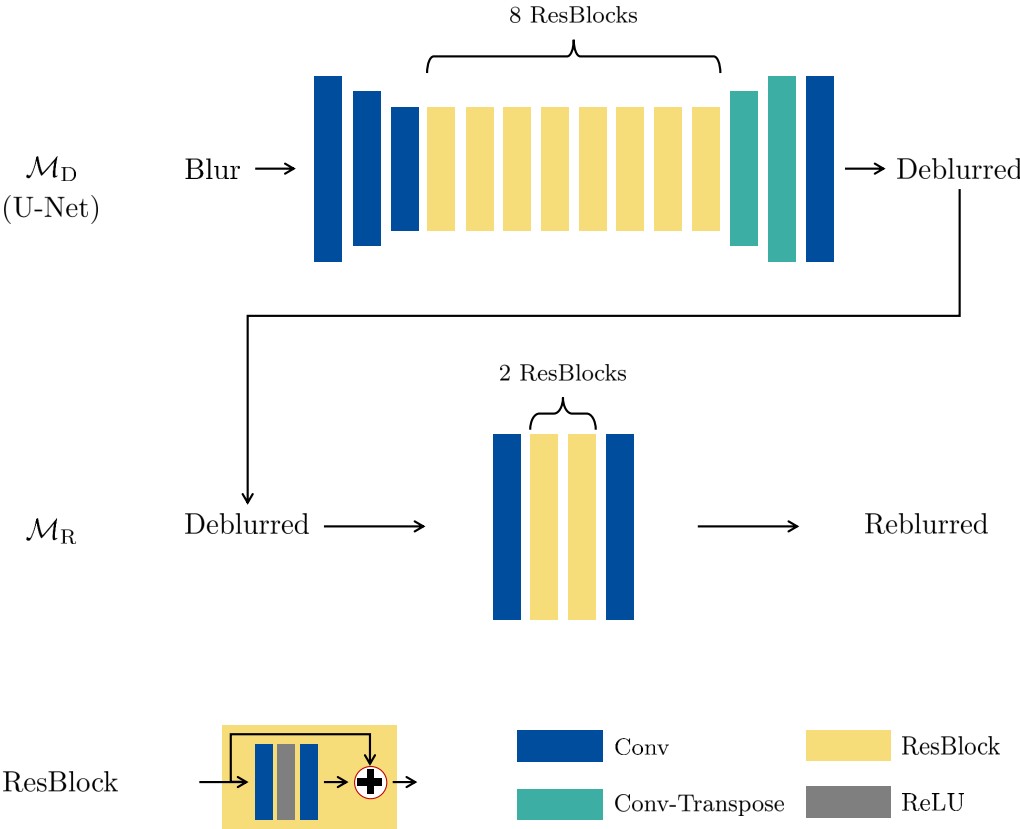

Figure A: **The baseline U-Net architecture and the reblurring module architecture** We use the same reblurring module for all experiments except the number of ResBlocks.
In addition to the U-Net, experiments were conducted with state-of-the-art deblurring models based on SRN (Tao et al., 2017) and DMPHN (Zhang et al., 2019). SRN (Tao et al., 2018) was originally designed to operate on grayscale images with a LSTM module. Later, the authors released the

| # | Layer description | Output shape |
|---|---|---|
| | Input | $3 \times H \times W$ |
| 1 | $5 \times 5$ conv | $64 \times H \times W$ |
| 2 | $3 \times 3$ conv | $128 \times H/2 \times W/2$ |
| 3 | $3 \times 3$ conv | $192 \times H/4 \times W/4$ |
| 4-19 | 8 ResBlocks ($3 \times 3$) | $192 \times H/4 \times W/4$ |
| 20 | $3 \times 3$ conv | $128 \times H/2 \times W/2$ |
| 21 | $3 \times 3$ conv | $64 \times H \times W$ |
| 22 | $5 \times 5$ conv | $3 \times H \times W$ |

Table A: **U-Net module specifics**

| Method | LPIPS$_\downarrow$ | NIQE$_\downarrow$ | PSNR$^\uparrow$ | SSIM$^\uparrow$ |
|---|---|---|---|---|
| DMPHN ($\mathcal{L}_1$ only) | 0.1184 | 5.542 | 31.42 | 0.9191 |
| DHN ($\mathcal{L}_1$ only) | **0.1179** | **5.490** | **31.53** | **0.9207** |

Table B: **DMPHN modification results on GOPRO dataset.** DHN without patch-wise convolution brings improved accuracy.

sRGB version code without LSTM, exhibiting an improved accuracy. We adopted the revised SRN structure in our experiments.

The other model we chose is based on DMPHN (1-2-4-8) (Zhang et al., 2019). DMPHN performs hierarchical residual refinement to produce the final output. The model consists of convolutional layers with ReLU activations that are spatially shift-equivariant. In Zhang et al. (2019), each level splits the given image and performs the convolutional operation on the divided patches. As the convolutional weights do not differ by the input patches, the operations do not necessarily have to be done patch-wise. Thus, we remove the multi-patch strategy and perform the convolution on the whole given input without dividing the image into patches. We refer to the modified model as DHN. As shown in Table B, convolution on the whole image compared with patch-wise convolution brings higher accuracy.

**Metrics.** To quantitatively compare the deblurred images in the following sections, we use PSNR, SSIM (Wang et al., 2004), LPIPS (Zhang et al., 2018b), and NIQE (Mittal et al., 2012). In the image deblurring literature, SSIM has been measured by MATLAB `ssim` function on sRGB images with $H \times W \times C$. SSIM was originally developed for grayscale images and MATLAB `ssim` function for a 3-dimensional tensor considers an image to be a 3D grayscale volume image. Thus, most of the previous SSIM measures were not accurate, leading to higher values. Instead, we measured all the SSIM for each channel separately and averaged them. We used `skimage.metrics.structural_similarity` function in the scikit-image package for python to measure SSIM for multi-channel images.

**Training.** For all the experiments, we performed the same training process for a fair comparison. On the GOPRO dataset (Nah et al., 2017), we trained each model for 4000 epochs. On the REDS dataset (Nah et al., 2019), the models are trained for 200 epochs. Adam (Kingma & Ba, 2014) optimizer is used in all cases. When calculating distance between images with Lp norm, we always set $p = 1$, using L1 distance. Starting from the initial learning rate $1 \times 10^{-4}$, the learning rate halves when training reaches 50%, 75%, and 90% of the total epochs. We used PyTorch 1.8.1 with CUDA 11.1 to implement the deblurring methods. Mixed-precision training (Micikevicius et al., 2017) is employed to accelerate operations on RTX 2080 Ti GPUs.

## C   DETERMINING THE REBLURRING MODULE SIZE

As our reblurring loss $\mathcal{L}_R$ is realized by $\mathcal{M}_R$, the reblurring module design plays an essential role. As shown in Figure A, the $\mathcal{M}_R$ architecture is a simple ResNet. Table D shows the relation between the deblurring performance and $\mathcal{M}_R$ size by changing the number of ResBlocks.

For all deblurring module $\mathcal{M}_D$ architectures, LPIPS was the best when the number of ResBlocks, $n = 2$. NIQE showed good performance when $2 \leq n \leq 3$. PSNR and SSIM had tendency to

| # | Layer description | Output shape |
|---|---|---|
| | Input | $3 \times H \times W$ |
| 1 | $5 \times 5$ conv | $64 \times H \times W$ |
| 2-5 | 2 ResBlocks ($5 \times 5$) | $64 \times H \times W$ |
| 6 | $5 \times 5$ conv | $3 \times H \times W$ |

Table C: **Reblurring module specifics**

| Method | LPIPS$_\downarrow$ | NIQE$_\downarrow$ | PSNR$^\uparrow$ | SSIM$^\uparrow$ |
|---|---|---|---|---|
| U-Net ($\mathcal{L}_1$ only) | 0.1635 | 5.996 | **29.66** | **0.8874** |
| $+\mathcal{L}_{\text{Reblur, n1}}$ | 0.1365 | 5.629 | 29.58 | 0.8869 |
| $+\mathcal{L}_{\text{Reblur, n2}}$ | **0.1238** | **5.124** | 29.44 | 0.8824 |
| $+\mathcal{L}_{\text{Reblur, n3}}$ | 0.1386 | 5.448 | 29.38 | 0.8819 |
| $+\mathcal{L}_{\text{Reblur, n4}}$ | 0.1415 | 5.513 | 29.25 | 0.8789 |
| SRN ($\mathcal{L}_1$ only) | 0.1246 | 5.252 | 30.62 | 0.9078 |
| $+\mathcal{L}_{\text{Reblur, n1}}$ | 0.1140 | 5.136 | **30.74** | **0.9104** |
| $+\mathcal{L}_{\text{Reblur, n2}}$ | **0.1037** | 4.887 | 30.57 | 0.9074 |
| $+\mathcal{L}_{\text{Reblur, n3}}$ | 0.1091 | **4.875** | 30.50 | 0.9060 |
| $+\mathcal{L}_{\text{Reblur, n4}}$ | 0.1155 | 5.041 | 30.53 | 0.9056 |
| DHN ($\mathcal{L}_1$ only) | 0.1179 | 5.490 | **31.53** | 0.9207 |
| $+\mathcal{L}_{\text{Reblur, n1}}$ | 0.0975 | 5.472 | **31.53** | **0.9217** |
| $+\mathcal{L}_{\text{Reblur, n2}}$ | **0.0837** | 5.076 | 31.34 | 0.9177 |
| $+\mathcal{L}_{\text{Reblur, n3}}$ | 0.0845 | **4.963** | 31.26 | 0.9159 |
| $+\mathcal{L}_{\text{Reblur, n4}}$ | 0.0861 | 5.041 | 31.19 | 0.9149 |

Table D: **The effect of reblurring loss on GOPRO dataset by the reblurrimg module size.** Reblurring module size varies by the number of ResBlocks.

decrease when $n \geq 1$. For larger number of ResBlocks, we witnessed sharper edges could be obtained but sometimes, cartoon artifacts with over-strong edges were witnessed.

Considering the trade-off between the PSNR and the perceptual metrics, we chose $n \in \{1, 2\}$ in the following experiments. $n = 1$ finds balance between the PSNR and LPIPS and $n = 2$ puts more weight on the perceptual quality.

## D  COMBINING REBLURRING LOSS WITH OTHER PERCEPTUAL LOSSES

Our reblurring loss is a new perceptual loss that is sensitive to blurriness of an image, a type of image structure-level information while other perceptual losses such as VGG loss (Johnson et al., 2016) and adversarial loss (Ledig et al., 2017) are more related to the high-level contexts. As VGG model (Simonyan & Zisserman, 2014) is trained to recognize image classes, optimizing with VGG loss could make an image better recognizable. In the GAN frameworks (Goodfellow et al., 2014), it is well known that discriminators can easily tell fake images from real images (Wang et al., 2020), being robust against JPEG compression and blurring. In the adversarial loss from the discriminator, the realism difference could be more salient than other features such as blurriness.

With the perceptual loss functions designed with different objectives, combining them could bring visual quality improvements in various aspects. Tables E and F show the effect of applying our reblurring loss jointly with other perceptual losses on GOPRO and REDS datasets. We omit the loss coefficients for simplicity. We used weight 0.3 for the VGG loss $\mathcal{L}_{\text{VGG}}$ and 0.001 for the adversarial loss, $\mathcal{L}_{\text{Adv}}$. We witness LPIPS and NIQE further improves when our reblurring loss is combined with $\mathcal{L}_{\text{VGG}}$ or $\mathcal{L}_{\text{Adv}}$.

| Method | LPIPS$_\downarrow$ | NIQE$_\downarrow$ | PSNR$^\uparrow$ | SSIM$^\uparrow$ |
|---|---|---|---|---|
| SRN ($\mathcal{L}_1$ only) | 0.1246 | 5.252 | 30.62 | 0.9078 |
| $+\mathcal{L}_{\text{VGG}}$ | 0.1037 | 4.945 | 30.60 | 0.9074 |
| $+\mathcal{L}_{\text{VGG}} + \mathcal{L}_{\text{Reblur, n2}}$ | **0.0928** | **4.671** | 30.64 | 0.9079 |
| $+\mathcal{L}_{\text{Adv}}$ | 0.1141 | 4.960 | 30.53 | 0.9068 |
| $+\mathcal{L}_{\text{Adv}} + \mathcal{L}_{\text{Reblur, n2}}$ | **0.1014** | **4.811** | 30.56 | 0.9075 |
| DHN ($\mathcal{L}_1$ only) | 0.1179 | 5.490 | 31.53 | 0.9207 |
| $+\mathcal{L}_{\text{VGG}}$ | 0.0994 | 5.022 | 31.48 | 0.9195 |
| $+\mathcal{L}_{\text{VGG}} + \mathcal{L}_{\text{Reblur, n2}}$ | **0.0773** | **4.897** | 31.28 | 0.9161 |
| $+\mathcal{L}_{\text{Adv}}$ | 0.0969 | 5.026 | 31.46 | 0.9188 |
| $+\mathcal{L}_{\text{Adv}} + \mathcal{L}_{\text{Reblur, n2}}$ | **0.0835** | **4.799** | 31.28 | 0.9162 |

Table E: **Results on GOPRO dataset by adding reblurring loss to the other preceptual losses.**

| Method | LPIPS$_\downarrow$ | NIQE$_\downarrow$ | PSNR$^\uparrow$ | SSIM$^\uparrow$ |
|---|---|---|---|---|
| SRN ($\mathcal{L}_1$ only) | 0.1148 | 3.392 | 31.89 | 0.8999 |
| $+\mathcal{L}_{\text{VGG}}$ | 0.1000 | 3.256 | 31.86 | 0.9001 |
| $+\mathcal{L}_{\text{VGG}} + \mathcal{L}_{\text{Reblur, n2}}$ | **0.0868** | **2.835** | 31.83 | 0.9015 |
| $+\mathcal{L}_{\text{Adv}}$ | 0.1158 | 3.395 | 31.84 | 0.8993 |
| $+\mathcal{L}_{\text{Adv}} + \mathcal{L}_{\text{Reblur, n2}}$ | **0.0934** | **2.836** | 32.00 | 0.9061 |
| DHN ($\mathcal{L}_1$ only) | 0.0942 | 3.288 | 32.65 | 0.9152 |
| $+\mathcal{L}_{\text{VGG}}$ | 0.0812 | 3.171 | 32.61 | 0.9146 |
| $+\mathcal{L}_{\text{VGG}} + \mathcal{L}_{\text{Reblur, n2}}$ | **0.0723** | **2.821** | 32.48 | 0.9133 |
| $+\mathcal{L}_{\text{Adv}}$ | 0.0956 | 3.218 | 32.58 | 0.9128 |
| $+\mathcal{L}_{\text{Adv}} + \mathcal{L}_{\text{Reblur, n2}}$ | **0.0820** | **2.809** | 32.45 | 0.9121 |

Table F: **Results on REDS dataset by adding reblurring loss to the other preceptual losses.**

## E    TEST-TIME ADAPTATION DETAILS

We describe the detailed self-supervised test-time adaptation process. At test time, the learning rate is set to $\mu = 3 \times 10^{-6}$. From the initial deblurring result $L^0$, the self-supervised loss is iteratively minimized by updating the weights of $\mathcal{M}_{\text{D}}$. As the self-supervised loss in equation 6 only cares about image sharpness, the image may have color drifting issues. Thus, finally, we match the histogram of the updated image $L^N$ to the histogram of $L^0$. The overall process is shown in Algorithm A.

---

**Algorithm A** Optimization process in test-time adaptation

---

1: **procedure** TEST-TIME ADAPTATION($B, \mathcal{M}_{\text{D}}, \mathcal{M}_{\text{R}}$)
2:     Test-time learning rate $\mu \leftarrow 3 \times 10^{-6}$.
3:     $\theta_{\text{D}} \leftarrow$ Weights of $\mathcal{M}_{\text{D}}$.
4:     $L^0 = \mathcal{M}_{\text{D}}(B)$.
5:     **for** $i = 0 \ldots N - 1$ **do**
6:         $L_*^i = \mathcal{M}_{\text{D}}(B)$.
7:         $\mathcal{L}_{\text{reblur}}^{\text{self}} = \|\mathcal{M}_{\text{R}}(\mathcal{M}_{\text{D}}(B)) - L_*^i\|$.
8:         Update $\theta_{\text{D}}$ by $\nabla_{\theta_{\text{D}}} \mathcal{L}_{\text{Reblur}}^{\text{self}}$ and $\mu$.
        $L^N = \mathcal{M}_{\text{D}}(B)$.
9:     $L_{\text{Adapted}}^N = \texttt{histogram\_matching}(L^N, L_*^0)$
10:    **return** $L_{\text{Adapted}}^N$

---

## F    PERCEPTION VS. DISTORTION TRADE-OFF

It is known in image restoration literature that the distortion error and the perceptual quality error are in trade-off relation (Blau & Michaeli, 2018; Blau et al., 2018). The relation is often witnessed by training a single model with different loss functions. In most cases, to obtain a better perceptual quality from a single model architecture, retraining with another loss from scratch is necessary. Our

test-time adaptation from self-supervised reblurring loss, in contrast, can provide the steps toward perceptual quality without full retraining.

In Figure B and C, we present the perception-distortion trade-off from our test-time adaption. LPIPS and NIQE scores consistently improve from each adaptation step in both SRN and DHN models. While PSNR is moderately sacrificed from the adaptation, SSIM improves in the early steps as it more reflects the structural information. Our results show improved trade-off between the distortion and perception metrics over the baseline models trained with L1 loss.

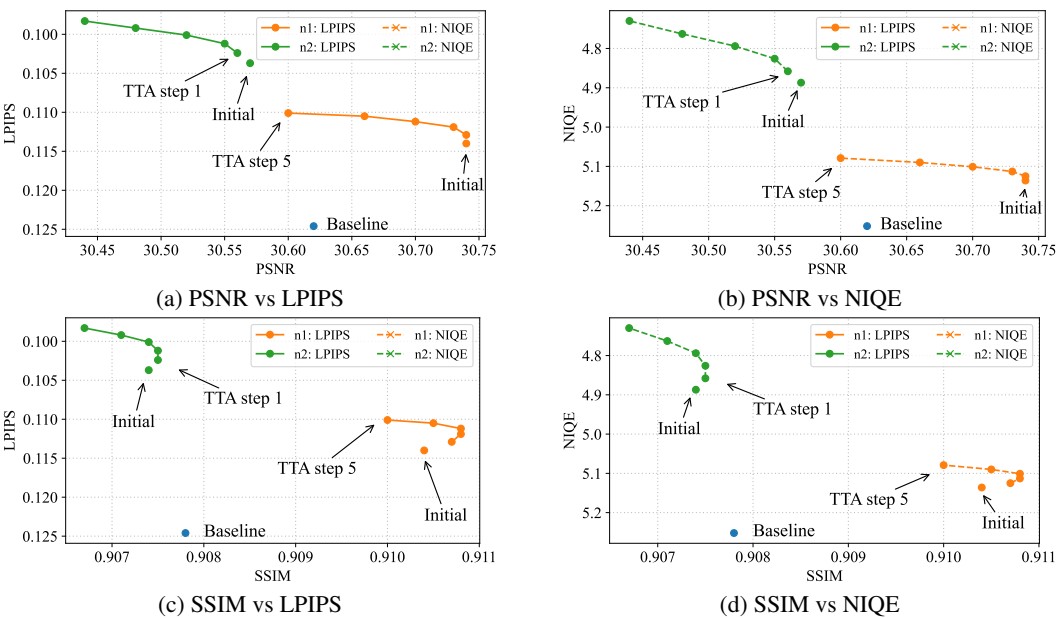

Figure B: **Perception-distortion trade-off from test-time adaptation applied to SRN model on GOPRO dataset.**

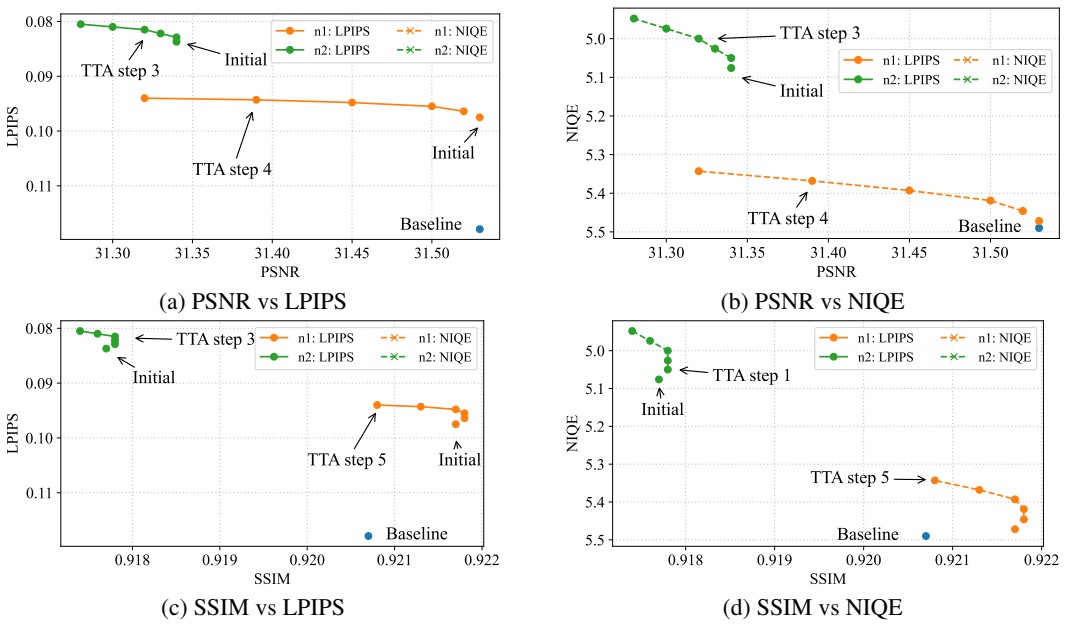

Figure C: **Perception-distortion trade-off from test-time adaptation applied to DHN model on GOPRO dataset.**

| Model | Optimization | On GOPRO dataset | | | | On REDS dataset | | | |
|---|---|---|---|---|---|---|---|---|---|
| | | LPIPS$_\downarrow$ | NIQE$_\downarrow$ | PSNR$^\uparrow$ | SSIM$^\uparrow$ | LPIPS$_\downarrow$ | NIQE$_\downarrow$ | PSNR$^\uparrow$ | SSIM$^\uparrow$ |
| SRN | $\mathcal{L}_1$ | 0.1246 | 5.252 | 30.62 | 0.9078 | 0.1148 | 3.392 | 31.89 | 0.8999 |
| | $\mathcal{L}_1 + \mathcal{L}_{\text{Reblur, n1}}$ | 0.1140 | 5.136 | 30.74 | 0.9104 | 0.1071 | 3.305 | 32.01 | 0.9044 |
| | + TTA step 5 | **0.1101** | 5.079 | 30.60 | 0.9100 | 0.1029 | 3.278 | 31.83 | 0.9040 |
| | + TTA step 10 | 0.1103 | 5.036 | 30.11 | 0.9048 | **0.1025** | **3.261** | 31.29 | 0.8996 |
| | + TTA step 20 | 0.1223 | 4.968 | 28.44 | 0.8806 | 0.1116 | 3.265 | 29.59 | 0.8807 |
| | + TTA step 30 | 0.1470 | **4.924** | 26.42 | 0.8411 | 0.1306 | 3.301 | 27.73 | 0.8523 |
| | $\mathcal{L}_1 + \mathcal{L}_{\text{Reblur, n2}}$ | 0.1037 | 4.887 | 30.57 | 0.9074 | 0.0947 | 2.875 | 31.82 | 0.9026 |
| | + TTA step 5 | 0.0983 | 4.730 | 30.44 | 0.9067 | **0.0909** | 2.798 | 31.50 | 0.9008 |
| | + TTA step 10 | **0.0962** | 4.569 | 30.07 | 0.9024 | 0.0913 | 2.741 | 30.87 | 0.8945 |
| | + TTA step 20 | 0.1021 | 4.274 | 28.83 | 0.8836 | 0.1033 | **2.699** | 29.09 | 0.8697 |
| | + TTA step 30 | 0.1199 | **4.045** | 27.26 | 0.8529 | 0.1259 | 2.729 | 27.20 | 0.8326 |

Table G: **Quantitative analysis of the reblurring losses and test-time adaptation applied to SRN on GOPRO and REDS datasets.**

## G    TEST-TIME ADAPTATION EFFECTS

In Table G, we quantitatively compare the deblurred results from test-time adaptation in terms of a no-reference metric, NIQE, and reference-based metrics, LPIPS, PSNR, and SSIMs. By performing TTA up to 30 steps as described in Algorithm A, we show that LPIPS and NIQE could be improved to a degree. On both GOPRO and REDS datasets, NIQE has a tendency to improve further after LPIPS has stopped its improvement. This is due to the self-supervised nature of test-time adaptation that considers image sharpness without reference.

In Figures D and E, we visually show the effect of test-time adaptation applied to SRN with a jointly trained reblurring module. By test-time adaptation, our model further improves the sharp edges of the images. In Figure D, the building structures and the horizontal lines are better witnessed. Also in Figure E, the vehicle's pole are better recovered and the text are clearer. While the PSNR and SSIM have decreased by test-time adaptation in Table G, perceptually, the results from test-time adaptation tend to be sharper.

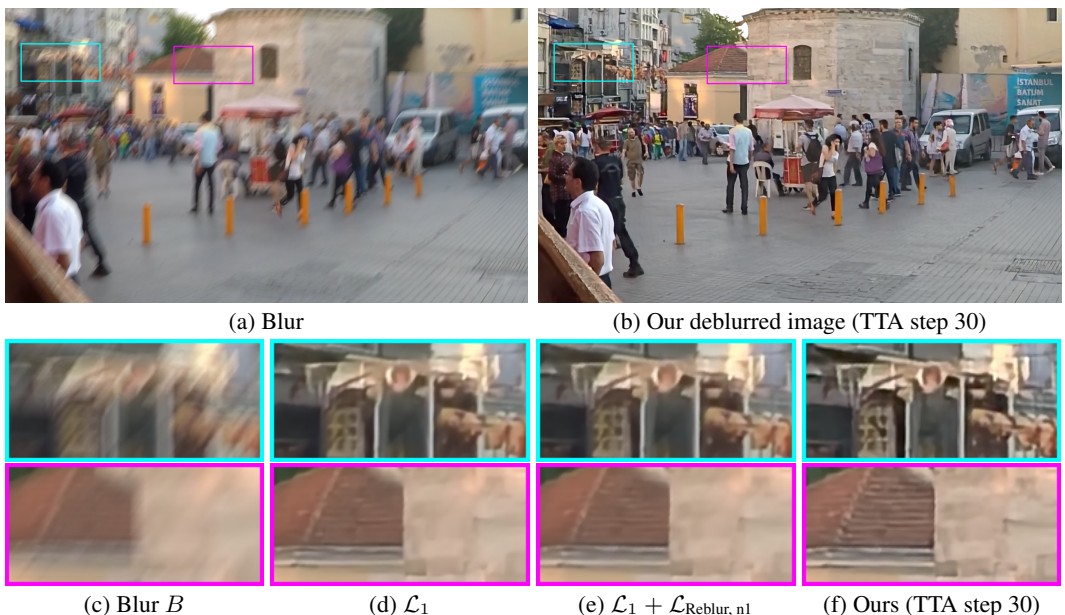

Figure D: **Visual comparison of deblurred results by reblurring loss and test-time adaptation on GOPRO dataset.**

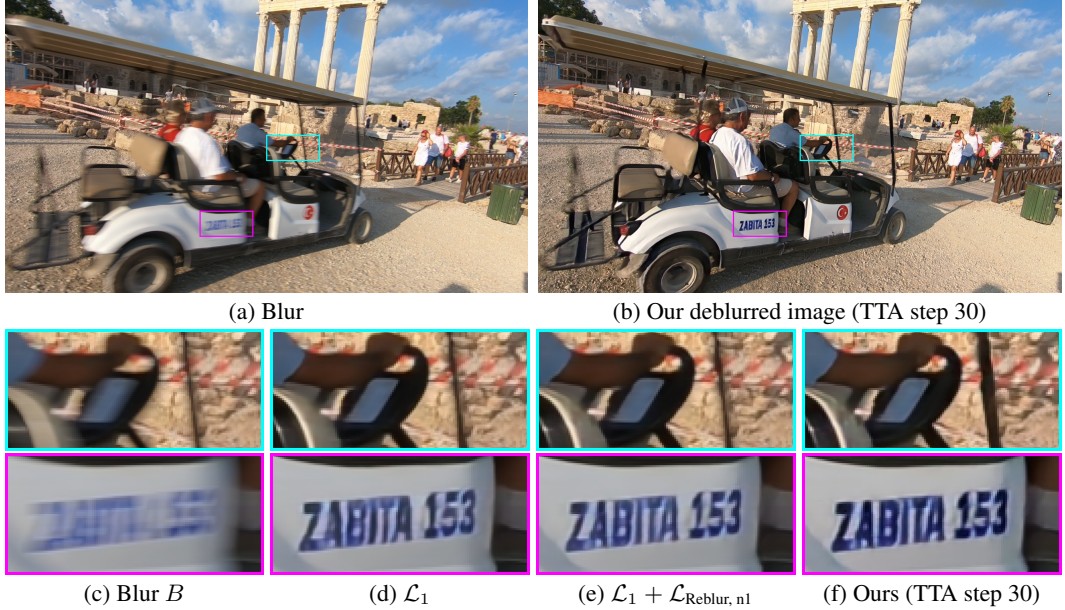

Figure E: **Visual comparison of deblurred results by reblurring loss and test-time adaptation on REDS dataset.**

