# OpenReview forum: "Clean Images are Hard to Reblur: Exploiting the Ill-Posed Inverse Task for Dynamic Scene Deblurring"
_ICLR.cc/2022/Conference — ICLR 2022 Poster_

### Official Review · Reviewer_oSPE · 2021-11-01

**Correctness:** 3
**Technical Novelty And Significance:** 3
**Empirical Novelty And Significance:** 3
**Recommendation:** 6
**Confidence:** 4

**Main Review:**

The main idea of the paper is based on an interesting observation. The proposed formulation/implementation seems to take advantage of the observation.

Strengths:
- The main idea of the paper is interesting and the proposed formulation seems to take advantage of the observation.
- Empirical results show that the method produces better results than the compared methods
The idea of using the re-blurring network to mitigate the out-of-distribution problems seems interesting.

Weaknesses:
- To implement the idea, the method introduces several loss terms that seem to be needed to avoid degenerate cases and other issues. In this sense, it feels that the formulation could be a little more elegant. The overall idea is that a deblurred image should be close to the target and also shouldn't have information about the original blur. In the end, the whole system seems to be doing this but I wonder if all the components are needed.

- The paper idea seems to be limited to deblurring. But in many parts the paper claims that the ideas are "applicable to general learning-based approaches" If this is one of the claims, there needs to be evidence supporting this.

- Analysis. There are many experiments but there's not too much analysis. From the exposition it is hard to claim that the method is actually improving over more classical losses like content loss (vgg loss, as shown in Table 4). So the question is more like if this method is better than simpler approaches? Also, if the issue is remaining blur, can't we just re-iterate the deblurring model till we reach a fixed-point?


Other comments:

- "inherent limitation of PSNR-oriented solutions" Do you mean the "regression-to-mean" problem that happens when we train with reference-based losses? This needs to be better discussed.

There are other reference based perceptual losses and relevant work that could be added to the overall discussion (not asking for more experiments but this could help improving the related work section):

Mechrez, R., Talmi, I., Shama, F. and Zelnik-Manor, L., 2018, December. Maintaining natural image statistics with the contextual loss. In Asian Conference on Computer Vision (pp. 427-443). Springer, Cham.

M. Delbracio, H. Talebei and P. Milanfar,  "Projected Distribution Loss for Image Enhancement," in 2021 IEEE International Conference on Computational Photography (ICCP), Haifa, Israel, 2021

Tariq, T., Tursun, O.T., Kim, M. and Didyk, P., 2020, August. Why Are Deep Representations Good Perceptual Quality Features?. In European Conference on Computer Vision (pp. 445-461). Springer, Cham.

Czolbe, S., Krause, O., Cox, I. and Igel, C., 2020. A Loss Function for Generative Neural Networks Based on Watson’s Perceptual Model. Advances in Neural Information Processing Systems, 33.

- Figure 1. This figure is unclear, please include an explanation for what is given on the top and bottom rows for each of the methods.

- "From a deblurred output, the reblurring module tries to make the reblurred image close to the original" So in practice this reblurring network is the inverse of the deblurring module. Why not present this module in this way?

- Writing is a little disorganized. For example in the introduction when referring to the "reblurring loss" it is unclear what is "the difference" that is mentioned. One of the listed contribution depends on this "reblurring loss" so this needs to be better explained in the introduction.

- The need for the pseudo-sharp image \hat{S}. The argument for using \hat{S} and not S seems to be rather empirical so this should be supported by evidence. In the current presentation there's a lot of emphasis on measuring sharpness but not realism, but then later for evaluating the quality perceptual metrics such as LPIPS and NIQE are used. So, there seems to be a mismatch between what is claimed and what is actually happening.

- The test-time adaptation inverts the re-blurring network. This needs more analysis in terms of:
What happens if we minimize this loss (gradient descent till convergence)? How many gradient descent steps are needed. This is not discussed much in the paper and I found it to be an interesting contribution. It will be also helpful to show cross-datasets performances (trained with GoPro and evaluated with a different dataset, e.g. HIDE, Real-Blur or REDS).

- Perceptual Distortion tradeoff. The perceptual distortion trade-off balances a reference based metric against a non-reference metric (e.g., distance to the natural image manifold). PSNR and LPIPS are both reference based metrics so it doesn't make sense (formally) to refer to this type of plot as a distortion-perception trade-off. Maybe the plot should be made with NIQE instead of LPIPS.


**Summary Of The Paper:**

This work addresses the problem of image deblurring by training a deep model in an end-to-end fashion. Similar to recent work, the method makes use of large paired (blurry,sharp) training datasets to train the deblurring network.

The main idea of the work is that a correctly deblurred image should not contain any information about the original blur. The method exploits this observation by using an auxiliary network that tries to re-blur the deblurred image so it has the same blur as the original one. If the deblurred image was perfectly deblurred this shouldn't be possible.

These two networks end up having a "similar" role as a generator-discriminator on a GAN set-up. But even if there's this superficial connection with GANs the formalism is completely different. Here the reblurring network outputs an image that is compared to another image (i.e., reference-based).

The work introduces three different loss terms: L_blur, L_sharp and L_reblur as long as the typical regression loss L1 between deblurred and sharp images.

- L_blur: is mainly used to train the re-blurring module;
- L_sharp: is used to avoid the re-blurring module to just apply the average blurring;
- L_reblur: is used as a regression loss on the space of re-blurred images.

Additionally, the paper proposes a test time adaptation where the reblurring network is used to boost the deblurring results on a given image. This is done by approximately inverting the re-blurring network using gradient descent.

The work presents several experiments including ablation studies with two of the popular (synthetic) datasets used in image/video deblurring (GoPro and REDS). Several comparisons with SOTA methods and some results on a real image dataset (Lai et al. 2016). Different quantitative metrics are given (PSNR,SSIM, LPIPS, NIQE) as well as several figures showing visual comparisons.

**Summary Of The Review:**

This paper introduces an interesting idea for improving image deblurring models: the deblurred image should not contain information/traces about the original blur.  This is a bold statement. The paper crystalizes this idea using two networks (deblurring and re-blurring network) that are co-trained with different loss terms. The implementation seems (a little) too complex with many terms that try to avoid different types of artifacts/degenerative cases. But in the end this seems like a valid implementation of the initial idea. The current analysis didn't convince me that the method is doing much better or better than other methods that just compare the restored and sharp images on a pre-computed feature space (is this a matter of tuning the contributions of each loss term?). I think the paper needs to better discuss and analyze this. Additionally, the idea seems to be specific for dealing with blur. Can the same idea apply to other (restoration) tasks?

---

> ### Author Response · Authors · 2021-11-14
> **Response to Reviewer oSPE**
>
> We thank reviewer **oSPE** for the detailed comments and suggestions.
> Please find our answers below.
>
> - The need for all components
>
> Our method indeed can work without sharpness the preservation loss term, too.
> However, as shown in Table 3, LPIPS and NIQE are better with sharpness preservation.
>
> - Applicability to general tasks
>
> We originally meant that our method is "applicable to general learning-based approaches **for deblurring**."
> We added 'for deblurring' for clarification.
>
> In principle, our method works as **blurring a clean image** is an ill-posed problem with many possible motion trajectories.
> Similarly, **noising a clean image** is another ill-posed problem as finding a specific noise is difficult.
> For example, there are many noise realizations under a white Gaussian noise assumption. (In practice, noise distribution has to be found, too.)
> Thus, we expect this method could be applied in denoising problems, too (clean images are hard to renoise) and considering it as future work.
>
> - Benefits of reblurring loss over other losses & iterative deblurring
>
> In Table 4, we show the effect of our *supervised reblurring loss without TTA* and other losses when PSNR and SSIM are at similar levels.
>
> Compared with VGG loss, *supervised reblurring loss* improves NIQE by -0.058 while having a decreased PSNR by -0.03 dB which is small considering the scales between NIQE and PSNR.
> Compared with adversarial loss, *supervised reblurring loss* improves all 4 metrics, LPIPS, NIQE, PSNR, SSIM.
>
> Also, in contrast to VGG loss and adversarial loss, our reblurring module enables self-supervised test-time adaptation.
> While GAN frameworks can try a similar approach with the discriminator, we didn't find good results in our experiments.
> Optimizing the deblurring module to fool discriminators at test-time caused several artifacts and content distortion.
>
> Also, iterative deblurring is also a valid idea as studied in (Park et al., 2020) by proposing a recurrent model architecture.
> On the other hand, in this work, we proposed loss functions to optimize a model rather than proposing a specific architecture.
> Loss function is an independent component of deep learning as well as the model architecture.
> Comparing the effect of a loss function and a model architecture is beyond the scope of this paper.
> Still, it is possible to use our reblurring loss to the iterative deblurring framework, too.
>
> - The expression, "inherent limitation of PSNR-oriented solutions"
>
> Yes, we mean the "regression-to-mean" problem and we thank **oSPE** for suggesting a simple and clear expression to improve the exposition.
> We changed the sentence as:
> "Still, most methods tend to suffer from the blurry predictions due to the regression-to-mean behavior often witnessed in ill-posed problems with large solution space."
>
> - Additional references
>
> We thank **oSPE** for the suggestion to make the related works section richer.
> We included the papers in our related works section.
>
> - Figure 1 clarity
>
> We changed captions and added arrows to clarify Figure 1.
>
> - Presenting reblurring as the inverse of deblurring
>
> We changed the sentence to:
> "From a deblurred output, the reblurring module performs the inverse operation of deblurring, trying to reconstruct the original blurry image."
>
> - Clarifying "the difference"
>
> We changed the sentence as:
> "We propose to use the difference between non-ideally deblurred image and the ideal sharp image in terms of reblurring feasibility
> as the new optimization objective, \emph{reblurring loss} for the image deblurring problem."
>
> - Need for pseudo-sharp image
>
> We used \hat{S} for training purposes, not for evaluation.
> As \hat{S} is an output from M_D that is trained by ourselves, measuring the fidelity of the deblurred results with \hat{S} is not very desirable.
> Especially, \hat{S} from every different experiment would be different as M_D is trained with different loss functions.
> Also, we would like to note that NIQE is a no-reference metric that does not use ground truth for evaluation.
>
> Instead, to provide clearer explanation, in Section 4.2, we added the following sentences:
> For neural networks, it is very easy to discriminate real and fake images (Wang et al., 2020).
> By using a fake image \hat{S} instead of a real image S, we let M_R focus on the sharpness of an image and avoid being distracted by a more obvious difference between real and fake images.
>
> - Test-time adaptation analysis
>
> In Figures 6, 11 and 12, we show the effect of test-time adaptation at each step.
> Each step improves LPIPS and NIQE while PSNR and SSIM is sacrificed, showing a typical trade-off between perception-distortion metrics. (Blau & Michaeli, 2018; Blau et al., 2018)
>
> We will show additional cross-dataset experiments later.
>
> - Perception-Distortion tradeoff
>
> In Figure 6, the dashed lines indicate NIQE, the corresponding y-axis shown on the right.
> In Appendix section F, the relation between NIQE with respect to PSNR and SSIM are shown.

---

> > ### Author Response · Authors · 2021-11-22
> > **Test-Time Adaptation Analysis (more steps & cross-dataset experiments)**
> >
> > We thank **oSPE** for suggesting valuable analysis.
> >
> > * Further TTA iterations
> >
> > We performed experiments with TTA for up to 30 steps on GOPRO dataset with SRN model.
> > For both reblurring module size 1 and 2, LPIPS, a reference-based metric, did not improve beyond a certain limit.
> > On the other hand, NIQE, a no-reference metric, continued to improve.
> > We witnessed oversharpening artifacts after many TTA steps which lead to deviation from the soft texture in GT scenes, however, with clearer edges.
> > This is due to our self-supervised reblurring loss (that enables TTA) focusing on the sharpness of an image without referring to GT.
> >
> > We visually show the effect of TTA at more steps in the Appendix, Figures 13 and 14.
> >
> > loss | TTA steps | LPIPS | NIQE | PSNR | SSIM | loss | TTA steps | LPIPS | NIQE | PSNR | SSIM
> > --- | :---: | --- | --- | --- | --- | --- | :---: | --- | --- | --- | ---
> > L1 (baseline) | - | **0.1246** | **5.252** | 30.62 | 0.9078 | L1 (baseline) | - | **0.1246** | **5.252** | 30.62 | 0.9078
> > L1 + Reblur (n1) | 0 | 0.1140 | 5.136 | 30.74 | 0.9104 | L1 + Reblur (n2) | 0 | 0.1037 | 4.887 | 30.57 | 0.9074
> > L1 + Reblur (n1) | 5 | **0.1101** | 5.079 | 30.60 | 0.9100 | L1 + Reblur (n2) | 5 | 0.0983 | 4.730 | 30.44 | 0.9067
> > L1 + Reblur (n1) | 10 | 0.1103 | 5.036 | 30.11 | 0.9048 | L1 + Reblur (n2) | 10 | **0.0962** | 4.569 | 30.07 | 0.9024
> > L1 + Reblur (n1) | 20 | 0.1223 | 4.968 | 28.44 | 0.8806 | L1 + Reblur (n2) | 20 | 0.1021 | 4.274 | 28.83 | 0.8836
> > L1 + Reblur (n1) | 30 | 0.1470 | **4.924** | 26.42 | 0.8411 | L1 + Reblur (n2) | 30 | 0.1199 | **4.045** | 27.26 | 0.8529
> >
> > * Cross-dataset adaptation
> >
> > We performed cross-dataset adaptation experiments with the SRN model by training on REDS and testing on GOPRO.
> > Similar to single-dataset experiments, LPIPS stopped improving after a certain point while NIQE consistently improved.
> >
> > loss | TTA steps | LPIPS | NIQE | PSNR | SSIM | loss | TTA steps | LPIPS | NIQE | PSNR | SSIM
> > --- | :---: | --- | --- | --- | --- | --- | :---: | --- | --- | --- | ---
> > L1 (baseline) | - | **0.1442** | **5.059** | 28.37 | 0.8796 | L1 (baseline) | - | **0.1442** | **5.059** | 28.37 | 0.8796
> > L1 + Reblur (n1) | 0 | **0.1386** | **4.991** | 28.48 | 0.8834 | L1 + Reblur (n2) | 0 | **0.1296** | **4.105** | 27.95 | 0.8763
> > L1 + Reblur (n1) | 1 | 0.1376 | 4.981 | 28.46 | 0.8843 | L1 + Reblur (n2) | 1 | 0.1291 | 4.074 | 27.91 | 0.8766
> > L1 + Reblur (n1) | 2 | 0.1369 | 4.967 | 28.42 | 0.8842 | L1 + Reblur (n2) | 2 | 0.1289 | 4.041 | 27.84 | 0.8759
> > L1 + Reblur (n1) | 3 | 0.1364 | 4.954 | 28.35 | 0.8837 | L1 + Reblur (n2) | 3 | **0.1290** | 4.009 | 27.76 | 0.8749
> > L1 + Reblur (n1) | 4 | 0.1359 | 4.943 | 28.29 | 0.8835 | L1 + Reblur (n2) | 4 | 0.1291 | 3.977 | 27.68 | 0.8740
> > L1 + Reblur (n1) | 5 | **0.1356** | **4.932** | 28.23 | 0.8833 | L1 + Reblur (n2) | 5 | 0.1295 | **3.946** | 27.60 | 0.8731

---

> > > ### Comment · Reviewer_oSPE · 2021-11-24
> > > **Final comments.**
> > >
> > > Thank you for the response.
> > >
> > > The authors' responses and the updated version of the manuscript helped to address some of the raised issues. I think this paper introduces and exploits a valid and interesting idea. Presentation and analysis has been improved in the revised version. Nonetheless, there's still room for improvement: there are multiple things presented: multiple loss terms, test-time adaptation, etc; and it's hard to make a consistent/full analysis with a clear understanding of all the pieces. I also tend to agree with reviewer Gxxe  observation *"A lot of the explanations in the paper are very hand-wavy, and not at all clear to the reader."*.
> > >
> > > To summarize. This work introduces an interesting idea exemplified (implemented) in a *reasonable* way. There could be other/better ways of exploiting this idea, but the idea itself (no remaining blur on restored image) is interesting. I'm advocating for acceptance of this paper.

---

> > > > ### Author Response · Authors · 2021-11-29
> > > > **Final response to oSPE**
> > > >
> > > > We thank **oSPE** for the thoughtful comments throughout the review and for acknowledging our idea to be valid and interesting.
> > > > We will do our best to further improve the exposition and provide better analysis and clarification on the role of each component.
> > > >
> > > > Best,
> > > > Authors

---

### Official Review · Reviewer_M9xc · 2021-11-02

**Correctness:** 4
**Technical Novelty And Significance:** 3
**Empirical Novelty And Significance:** 3
**Recommendation:** 8
**Confidence:** 4

**Main Review:**

1. Technical novelty
The observation that the "clean images are hard to reblur" is quite novel.
Following the observation, the proposed reblurring loss is logical and experimental results are consistent.
I'm concerned that the concept of deblur-reblur is similar to the GANs, but the authors well addressed the issue in the introduction.

2. Writing
Overall it is well-written and easy to read. However, more proofreading and re-arranging figures are needed before publication.
In Figure 1, the sharp image (supposed to be ground truth) and deblurred images are located in the top row. It may be a reasonable arrangement since they are seemingly similar. However since they have different meanings, there should be a better arrangement for easier understanding.
Readers may be familiar with the blurry and deblurred image, but not the reblurred image since it is quite a new concept. So additional figures explaining and emphasizing such concepts would help the understanding.
In Figure 2, there are B and S in both of Figure 2 (a) and (b), where the arrangement is somewhat confusing.
Table 1 is mentioned just below the equation (1) but there is no explanation about M_D before Table 1.

3. Experimental results
Experimental results improve SOTA in perceptual measures such as LPIPS and NIQE. However, it did not improve PSNR and SSIM, which gives the impression that the deblurred image may distort the original image and obtain better perceptual measures by sharpening.

**Summary Of The Paper:**

A novel deblurring method is proposed by introducing the concept of reblurring loss. Conventional techniques deblur blurry images to some extent however there are some unremoved blur contents, where it can be used to reconstruct (reblur) blurry images. On the other hand, cleanly deblurred images have only sharp contents and it is difficult to reblur the images since no blurry cues are left. By using such observation, a reblurring module and a deblurring module with reblurring loss are proposed. Experimental results show that the proposed framework outperforms SOTA quantitatively and qualitatively on the GOPRO and REDS dataset.

**Summary Of The Review:**

The paper introduces an interesting observation and proposes novel modules based on the observation where the development is very logical. Overall, the paper is easy to read, however more proofreading is required. Experimental results improve the SOTA but there should be justification for some results.

---

> ### Author Response · Authors · 2021-11-14
> **Response to Reviewer M9xc**
>
> We thank **M9xc** for the valuable comments to improve the paper and exposition.
> Here are our answers.
>
> - Figure 1.
>
> We changed Figure 1 for clarification.
>
> We marked the blur and sharp images with bold text with the word 'True.'
> We changed the label of our deblurred image from 'Ours' to 'Ours (Deblurred)'
> We added arrows between the deblurred and the reblurred images to show that reblurring is generated from the deblurred images.
>
> - Figure 2.
>
> To avoid confusion, we put a vertical line between (a) and (b).
> We also put a more detailed description in the caption.
>
> - Table 1.
>
> We now refer to the deblurring module as: deblurring module $\mathcal{M}_{\text{D}}$
>
> - Perception-distortion trade-off
>
> Our method focuses on image sharpness by proposing a new type of perceptual loss.
> It is typical that applying perceptual losses essentially brings the gains in the perceptual metrics at the cost of PSNR.
> Such perception-distortion trade-off relation has been witnessed in general image restoration literature (Blau & Michaeli, 2018; Blau et al., 2018).
>
> Thus, our goal is to improve the trade-off between perception and distortion.
> In Table 4, we show that LPIPS and NIQE is better than other perceptual losses when PSNR and SSIM are in similar levels.
>
> Also, in Appendix Table 8, we show that for SRN and DHN, when the reblurring module is small (1 ResBlock), PSNR and SSIM remain at a similar level or improve.
> It means that while the trade-off relation essentially cannot be avoided, however, our trade-off itself is improved from the baseline.

---

> > ### Comment · Reviewer_M9xc · 2021-11-30
> > **Thanks for the rebuttal**
> >
> > All concerns in the review are well-address in the revised version, including adjusting figures and proofreading!

---

### Official Review · Reviewer_Gxxe · 2021-11-03

**Correctness:** 2
**Technical Novelty And Significance:** 3
**Empirical Novelty And Significance:** Not applicable
**Recommendation:** 5
**Confidence:** 4

**Main Review:**

This is an interesting idea. For the task of deblurring, it represents a logical and novel formulation of the “adversary”: rather than just saying whether the input is truly sharp or the output of a deblurring network, the reblurring network has to reproduce the original blurry image in the latter case.

While the results are encouraging, there are a number of issues with the paper:

The paper needs to do a much better job in clearly and systematically establishing the benefit of the proposed method against a standard adversarial loss and other perceptual losses (which are much simpler and more straightforward than the proposed reblurring loss). In particular, the efficacy of the network WITHOUT test-time adaptation should be established. This is because TTA is expensive, and it is not clear whether other methods couldn’t also be improved with TTA (for example, those based on standard adversarial losses) or by just having slower but deeper networks.

Table 4 seems to suggest minimal additional benefit of the reblur loss (without TTA). There is some benefit in NIQE, none in LPIPS, and PSNR and SSIM get worse. This comparison is also to a specific weight on the adversarial loss. Did the authors sweep for different loss weights to find the optimal one for training only with an adversarial (+ VGG) loss? This is what I mean by a systematic evaluation: the paper should find the optimal hyperparameters for training with only a regular adversarial loss and compare to the optimal setting for the proposed method — without TTA and using a comparable architecture for the discriminator as the proposed MR.


The paper also needs more comprehensive results reported for comparisons to state-of-the-art. There are no quantitative results, and the qualitative results are only with TTA. Visually, the improvement of the proposed method with TTA over DeblurGAN is relatively small. It would be natural to wonder if that improvement disappears without TTA, and if infact, the proposed method does worse than DeblurGAN. As noted, TTA is expensive, and an orthogonal approach that could potentially be applied to other adversarially trained deblurring networks.

The paper, especially the introduction, could be a lot clearer in explaining the basic idea. Figure 1 is great in terms of layout, but the captions / legend is not informative making it very unintuitive.  The caption should at the very least explain what L, S, \hat{S}, and B are without having readers to dig through the text in section 3. The “shared” is confusing — one would anyway assume that all M_R’s are the same, but the fact there’s shared lines only among pairs on the left and right makes us question if the two pairs are different from each other. Having (a) and (b) in Figure 1 with the corresponding headings is also confusing. They each respectively describe the training loss for M_R and M_D, and if I understand correctly, each iteration involves doing an update step of each. This does not come across from the figure. And finally, some more details of the test-time adaptation should be included in the main paper. Most of the text in the main paper in that section gives “intuition” — but is very confusing to read since the reader doesn’t know exactly what is being optimized and what is being updated (image or network weights).


A lot of the explanations in the paper are very hand-wavy, and not at all clear to the reader. This is especially true about the use of a pseudo sharp image instead of a sharp image: for example, I don’t understand what the authors mean by “In contrast to S that differ from the deblurred image L by the realness, \hat{S} can avoid being MR distracted by such an unintentional difference, focusing on image sharpness.” I’m actually not sure why using S-hat doesn’t just cause a degenerate solution where MD produces oversmoothed outputs all the time to prevent MR from figuring out if the original image was blurry or not.

**Summary Of The Paper:**

The paper proposes a novel approach to using deep networks for image deblurring. Since training with simply reconstruction losses usually leads to oversmoothed results, recent works have looked at using perceptual and adversarial losses.

The paper proposes an approach similar to an adversarial loss. If an image is not de-blurred entirely, it leaves within itself some tell-tale signs of the original blur. The paper proposes learning a “reblurring” network that given a “deblurred” image should yield the original “blurry” image, while given a true sharp image should output the image itself. The deblurring network is then trained in a sense to fool the re-blurring network, by generating images that would be left untouched by the reblurring network. The paper also proposes a “test time adaptation” approach.

**Summary Of The Review:**

Given the issues with the experiments and presentation above, I don't think the paper is ready for publication.

### Post-rebuttal

Increasing score from reject to borderline/marginal  reject. I think the paper still needs a bit more work, but would not object to it being accepted. Please see detailed comment in response to rebuttal below.

---

> ### Author Response · Authors · 2021-11-14
> **Response to Reviewer Gxxe**
>
> We thank **Gxxe** for finding our idea to be interesting and providing valuable comments.
> We tried our best to answer the questions.
>
> - Benefits and efficacy of reblurring loss without test-time adaptation
>
> While we show and compare the effect on the quantitative metrics **without test-time adaptation** in Table 4, we note that the **different perceptual losses (VGG loss, adversarial loss, and our reblurring loss) play different roles and do not compete with each other**.
> For example, adversarial loss makes image realistic, and our reblurring loss focuses on reducing motion blur.
> Thus, we propose our reblurring loss as a new perceptual loss with different behavior from the others.
> Reblurring loss can work together with the other loss terms rather than replacing them.
> In Appendix Tables 9 and 10, we show the effect of the joint perceptual loss terms further improves LPIPS and NIQE.
>
> We would like to note that test-time adaptation is possible from our reblurring module, but not very easily with adversarial loss.
> We tried TTA with adversarial loss in a similar manner as ours (by optimizing the deblurring module to fool discriminator).
> In our experiments, we did not find any good results.
> Test-time adaptation with discriminator caused various artifacts, content distortion, etc.
> We believe test-time adaptation with adversarial loss is another research topic to be studied but beyond the scope of this work.
>
> Deblurring quality improves with deeper networks, however, if optimized with L1, VGG, or adversarial loss terms, they essentially leave the blur footprint behind.
> In Figure 1, we show that heavier network (SE-Sharing) or the model trained with VGG and adversarial loss (DeblurGANv2) do not remove blur completely.
>
> - Table 4 results
>
> In Table 4, we compared the reblurring loss with VGG loss and adversarial loss.
> We chose the loss weights so that the PSNR and SSIM are in a similar level to compare the effect on LPIPS and NIQE.
> Compared with VGG loss, reblurring loss improves NIQE by -0.058 while having a decreased PSNR by -0.03 dB which is small considering the scales between NIQE and PSNR.
> Compared with adversarial loss, reblurring loss improves all 4 metrics, LPIPS, NIQE, PSNR, SSIM.
>
> We will report additional results with different loss weights.
>
> Here are the number of parameters used to construct each perceptual loss:
>
> type | #params
> :--- | ---:
> VGG | 20,024,384
> Discriminator | 1,616,224
> Reblurring module (2 Resblocks) | 419,200
>
> - Comparisons
>
> As our contribution is the way to optimize deblurring networks, we do not intend to directly compare our result with the other architecture's results.
> We presented quantitative comparisons on various architectures (UNet, SRN, DHN) by applying our loss function to them.
> In Figure 7, we show that our 1) the car plate numbers are much cleaner and 2) the pinkle and the ring fingers are better separated than the deblurred image of DeblurGAN-v2.
>
> We primarily compare the effect of the adversarial loss (major loss function of DeblurGAN-v2) and the **reblurring loss without TTA** in Figure 4. It shows that c-pillar of the car and the red bar is better recovered by reblurring loss without TTA.
>
> - Introduction
>
> We changed the captions of Figure 1 and added arrows.
> In Figure 2, we included explanation of each term and the overall behavior.
> We removed the 'shared' mark and the dashed lines.
>
> - Test-time adaptation explanation
>
> In Section 3.3 page 4, we put a new sentence to clarify that the model weights are being updated:
> We optimize the weights of M_D with fixed M_R.
>
> We added the pointer to Algorithm 1 in the Appendix by changing the sentence:
> please refer to the Appendix -> please refer to the Appendix Algorithm 1.
>
> - Pseudo-sharp image
>
> For neural networks, **discriminating a fake image from a real image is very easy** (Wang et al., 2020).
> In blur reconstruction loss (eq 1), we used L=M_D(B), **a fake image** generated by M_D, as an input to M_R.
> In sharpness preservation loss (eq 2), we use a pseudo-sharp image \hat{S}=M_D(S), **another fake image**, as an input to M_R.
> We intend to use both loss terms to let M_R learn to behave differently **by finding the blurriness of an image, not by discriminating real and sharp images**.
> If we use a real image S in (eq 2), M_R could learn to discriminate L and S as a fake and a real image without having to check the blurriness.
>
> Therefore, we updated Section 4.2 in our main manuscript as follows, for clarification:
> For neural networks, it is very easy to discriminate real and fake images (Wang et al., 2020).
> By using a fake image \hat{S} instead of a real image S, we let M_R focus on the sharpness of an image and avoid being distracted by a more obvious difference between real and fake images.
>
> \hat{S} does not cause a degenerate solution as M_D learns to deblur a blurry image.
> If \hat{S} is oversmoothed (blurry), then L would be even more blurrier than B and that will be penalized by the L1 loss function, |L - S|.

---

> > ### Author Response · Authors · 2021-11-21
> > **Perceptual loss weight sweep experiment (extending Table 4)**
> >
> > We further report the extended results for Table 4 with additional loss weight experiments.
> > As we remarked before, we compared the effects on LPIPS and NIQE of each perceptual loss when PSNR and SSIM are at similar levels.
> >
> > **Bold** numbers are the original numbers reported in Table 4, **_bold italic_** numbers are for the adversarial loss with a higher PSNR and SSIM.
> > As shown below, reblurring loss provides better LPIPS and NIQE than VGG loss or adversarial loss.
> > Still, we would like to note that different loss terms are not essentially competing with each other and could be used jointly.
> > Please refer to Tables 9 and 10 in the Appendix for the effect of combined loss terms.
> >
> > model architecture: SRN
> >
> > loss | loss weight ($\lambda$) | LPIPS | NIQE | PSNR | SSIM
> > --- | :---: | --- | --- | --- | --- |
> > L1 | - | **0.1246** | **5.252** | **30.62** | **0.9078**
> > L1 + $\lambda$ VGG | 0.1 | 0.1133 | 5.151 | 30.55 | 0.9070
> > L1 + $\lambda$ VGG | 0.3 | **0.1037** | **4.945** | **30.60** | **0.9074**
> > L1 + $\lambda$ VGG | 0.5 | 0.1030 | 4.903 | 30.56 | 0.9066
> > L1 + $\lambda$ VGG | 1.0 | 0.1000 | 4.819 | 30.45 | 0.9044
> > L1 + $\lambda$ Adv | 0.0001 | 0.1239 | 5.264 | 30.62 | 0.9078
> > L1 + $\lambda$ Adv | 0.0003 | **_0.1183_** | **_5.146_** | **_30.60_** | **_0.9079_**
> > L1 + $\lambda$ Adv | 0.001 | **0.1141** | **4.960** | **30.53** | **0.9068**
> > L1 + $\lambda$ Adv | 0.003 | 0.1123 | 4.877 | 30.52 | 0.9066
> > L1 + $\lambda$ Adv | 0.01 | 0.1040 | 4.605 | 30.50 | 0.9052
> > L1 + $\lambda$ Reblur (n1) | 1.0 | 0.1140 | 5.136 | 30.74 | 0.9104
> > L1 + $\lambda$ Reblur (n1) | 0.5 | 0.1138 | 5.158 | 30.68 | 0.9094
> > L1 + $\lambda$ Reblur (n2) | 1.0 | **0.1037** | **4.887** | **30.57** | **0.9074**
> > L1 + $\lambda$ Reblur (n2) | 0.5 | *0.1034* | *4.772* | *30.55* | *0.9075*

---

### Decision · Program_Chairs · 2022-01-20

**Decision:**

Accept (Poster)

**Comment:**

The paper introduces an idea that was found interesting by all reviewers (including Gxxe who recommends a marginal reject). A majority of the reviewers also point out a few weaknesses of the paper, notably in terms of clarity of several statements that were found to be hand-wavy (see the reviews of Gxxe and oSPE for more precise details). The area chair agrees with those statements, but overall, the originality of the idea introduced in this paper outweighs these weaknesses, and the experimental study is conducted in a reasonably convincing manner.

Even though there is room for improvements, the area chair is happy to recommend an accept, but encourages the authors to follow the constructive feedback provided by the reviewers for the camera-ready version.